# AdaBelief Optimizer: Adapting Stepsizes by the Belief in Observed Gradients

## Abstract

Optimization is at the core of modern deep learning. We propose AdaBelief optimizer to simultaneously achieve three goals: fast convergence as in adaptive methods, good generalization as in SGD, and training stability. The intuition for AdaBelief is to adapt the stepsize according to the "belief" in the current gradient direction. Viewing the exponential moving average (EMA) of the noisy gradient as the prediction of the gradient at the next time step, if the observed gradient greatly deviates from the prediction, we distrust the current observation and take a small step; if the observed gradient is close to the prediction, we trust it and take a large step. We validate AdaBelief in extensive experiments, showing that it outperforms other methods with fast convergence and high accuracy on image classification and language modeling. Specifically, on ImageNet, AdaBelief achieves comparable accuracy to SGD. Furthermore, in the training of a GAN on Cifar10, AdaBelief demonstrates high stability and improves the quality of generated samples compared to a well-tuned Adam optimizer.

## 1 Introduction

Modern neural networks are typically trained with first-order gradient methods, which can be broadly categorized into two branches: the accelerated stochastic gradient descent (SGD) family [1], such as Nesterov accelerated gradient (NAG) [2], SGD with momentum [3] and heavy-ball method (HB) [4]; and the adaptive learning rate methods, such as Adagrad [5], AdaDelta [6], RMSProp [7] and Adam [8]. SGD methods use a global learning rate for all parameters, while adaptive methods compute an individual learning rate for each parameter.

Compared to the SGD family, adaptive methods typically converge fast in the early training phases, but have poor generalization performance [9, 10]. Recent progress tries to combine the benefits of both, such as switching from Adam to SGD either with a hard schedule as in SWATS [11], or with a smooth transition as in AdaBound [12]. Other modifications of Adam are also proposed: AMSGrad [13] fixes the error in convergence analysis of Adam, Yogi [14] considers the effect of minibatch size, MSVAG [15] dissects Adam as sign update and magnitude scaling, RAdam [16] rectifies the variance of learning rate, Fromage [17] controls the distance in the function space, and AdamW [18] decouples weight decay from gradient descent. Although these modifications achieve better accuracy compared to Adam, their generalization performance is typically worse than SGD on large-scale datasets such as ImageNet [19]; furthermore, compared with Adam, many optimizers are empirically unstable when training generative adversarial networks (GAN) [20].

To solve the problems above, we propose "AdaBelief", which can be easily modified from Adam. Denote the observed gradient at step $t$ as $g_t$ and its exponential moving average (EMA) as $m_t$. Denote the EMA of $g_t^2$ and $(g_t - m_t)^2$ as $v_t$ and $s_t$, respectively. $m_t$ is divided by $\sqrt{v_t}$ in Adam, while it is divided by $\sqrt{s_t}$ in AdaBelief. Intuitively, $\frac{1}{\sqrt{s_t}}$ is the "belief" in the observation: viewing $m_t$ as

Submitted to 34th Conference on Neural Information Processing Systems (NeurIPS 2020). Do not distribute.

the prediction of the gradient, if $g_t$ deviates much from $m_t$, we have weak belief in $g_t$, and take a small step; if $g_t$ is close to the prediction $m_t$, we have a strong belief in $g_t$, and take a large step. We validate the performance of AdaBelief with extensive experiments.

## 2  Methods

### 2.1  Details of AdaBelief Optimizer

**Notations**   By the convention in [8], we use the following notations:

- $f(\theta) \in \mathbb{R}, \theta \in \mathbb{R}^d$: $f$ is the loss function to minimize, $\theta$ is the parameter in $\mathbb{R}^d$
- $\prod_{\mathcal{F},M}(y) = \arg\min_{x \in \mathcal{F}} ||M^{1/2}(x-y)||$: projection of $y$ onto a convex feasible set $\mathcal{F}$
- $g_t$: the gradient and step $t$
- $m_t$: exponential moving average (EMA) of $g_t$
- $v_t, s_t$: $v_t$ is the EMA of $g_t^2$, $s_t$ is the EMA of $(g_t - m_t)^2$
- $\alpha, \epsilon$: $\alpha$ is the learning rate, default is $10^{-3}$; $\epsilon$ is a small number, typically set as $10^{-8}$
- $\beta_1, \beta_2$: smoothing parameters, typical values are $\beta_1 = 0.9, \beta_2 = 0.999$
- $\beta_{1t}, \beta_{2t}$ are the momentum for $m_t$ and $v_t$ respectively at step $t$, and typically set as constant (e.g. $\beta_{1t} = \beta_1, \beta_{2t} = \beta_2, \forall t \in \{1, 2, ...T\}$

| **Algorithm 1:** Adam Optimizer | **Algorithm 2:** AdaBelief Optimizer |
|---|---|
| **Initialize** $\theta_0, m_0 \leftarrow 0 , v_0 \leftarrow 0, t \leftarrow 0$ | **Initialize** $\theta_0, m_0 \leftarrow 0 , s_0 \leftarrow 0, t \leftarrow 0$ |
| **While** $\theta_t$ not converged | **While** $\theta_t$ not converged |
| $\quad t \leftarrow t + 1$ | $\quad t \leftarrow t + 1$ |
| $\quad g_t \leftarrow \nabla_\theta f_t(\theta_{t-1})$ | $\quad g_t \leftarrow \nabla_\theta f_t(\theta_{t-1})$ |
| $\quad m_t \leftarrow \beta_1 m_{t-1} + (1 - \beta_1)g_t$ | $\quad m_t \leftarrow \beta_1 m_{t-1} + (1 - \beta_1)g_t$ |
| $\quad v_t \leftarrow \beta_2 v_{t-1} + (1 - \beta_2)g_t^2$ | $\quad s_t \leftarrow \beta_2 s_{t-1} + (1 - \beta_2)(g_t - m_t)^2$ |
| $\quad$ **Update** | $\quad$ **Update** |
| $\qquad \theta_t \leftarrow \prod_{\mathcal{F}, \sqrt{v_t}}\left(\theta_{t-1} - \frac{\alpha m_t}{\sqrt{v_t}+\epsilon}\right)$ | $\qquad \theta_t \leftarrow \prod_{\mathcal{F}, \sqrt{s_t}}\left(\theta_{t-1} - \frac{\alpha m_t}{\sqrt{s_t}+\epsilon}\right)$ |

**Comparison with Adam**   Adam and AdaBelief are summarized in Algo. 1 and Algo. 2, where all operations are element-wise, with differences marked in blue. Note that no extra parameters are introduced in AdaBelief. For simplicity, we omit the bias correction step. A detailed version of AdaBelief is in Appendix A. Specifically, in Adam, the update direction is $m_t/\sqrt{v_t}$, where $v_t$ is the EMA of $g_t^2$; in AdaBelief, the update direction is $m_t/\sqrt{s_t}$, where $s_t$ is the EMA of $(g_t - m_t)^2$. Intuitively, viewing $m_t$ as the prediction of $g_t$, AdaBelief takes a large step when observation $g_t$ is close to prediction $m_t$, and a small step when the observation greatly deviates from the prediction.

### 2.2  Intuitive explanation for benefits of AdaBelief

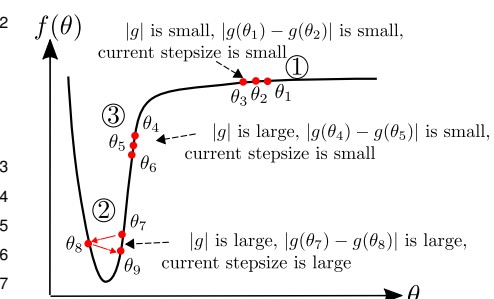

$f(\theta)$

$|g|$ is small, $|g(\theta_1) - g(\theta_2)|$ is small, current stepsize is small

$|g|$ is large, $|g(\theta_4) - g(\theta_5)|$ is small, current stepsize is small

$|g|$ is large, $|g(\theta_7) - g(\theta_8)|$ is large, current stepsize is large

Figure 1: An ideal optimizer considers curvature of the loss function, instead of taking a large (small) step where the gradient is large (small)

**AdaBelief uses curvature information**   Update formulas for SGD, Adam and AdaBelief are:

$$\Delta\theta_t^{SGD} = -\alpha m_t, \quad \Delta\theta_t^{Adam} = -\alpha m_t/\sqrt{v_t},$$
$$\Delta\theta_t^{AdaBelief} = -\alpha m_t/\sqrt{s_t} \tag{1}$$

Note that we name $\alpha$ as the "learning rate" and $|\Delta\theta_t^i|$ as the "stepsize" for the $i$th parameter. With a 1D example in Fig. 1, we demonstrate that AdaBelief uses the curvature of loss functions to improve training, with a detailed description below:

(1) In region ① in Fig. 1, the loss function is flat, hence the gradient is close to 0. In this case, an ideal optimizer should take a large stepsize. The stepsize of SGD is proportional to the EMA of the gradient, hence is small in this case; while both Adam and AdaBelief take a large stepsize, because the denominator ($\sqrt{v_t}$ and $\sqrt{s_t}$) is a small value.

74   (2) In region ②, the algorithm oscillates in a "steep and narrow" valley, hence both $|g_t|$ and $|g_t - g_{t-1}|$
75   is large. An ideal optimizer should decrease its stepsize, while SGD takes a large step (proportional
76   to $m_t$). Adam and AdaBelief take a small step because the denominator ($\sqrt{s_t}$ and $\sqrt{v_t}$) is large.

77   (3) In region ③, we demonstrate AdaBelief's advantage over Adam in the "large gradient, small
78   curvature" case. In this case, $|g_t|$ and $v_t$ are large, but $|g_t - g_{t-1}|$ and $s_t$ are small; this could happen
79   because of a small learning rate $\alpha$. In this case, an ideal optimizer should increase its stepsize. SGD
80   uses a large stepsize ($\sim \alpha|g_t|$); in Adam, the denominator $\sqrt{v_t}$ is large, hence the stepsize is small; in
81   AdaBelief, denominator $\sqrt{s_t}$ is small, hence the stepsize is large as in an ideal optimizer.

82   To sum up, AdaBelief scales the update direction by the change in gradient, which is related to the
83   Hessian. Therefore, AdaBelief considers curvature information and performs better than Adam.

84   **AdaBelief considers the sign of gradient in denominator**   We show the advantages of AdaBelief
85   with a 2D example in this section, which gives us more intuition for high dimensional cases. In Fig. 2,
86   we consider the loss function: $f(x, y) = |x| + |y|$. Note that in this simple problem, the gradient in
87   each axis can only take $\{1, -1\}$. Suppose the start point is near the $x-$axis, e.g. $y_0 \approx 0, x_0 \ll 0$.
88   Optimizers will oscillate in the $y$ direction, and keep increasing in the $x$ direction.
89   Suppose the algorithm runs for a long time ($t$ is large), so the bias of EMA ($\beta_1^t \mathbb{E}g_t$) is small:

$$m_t = EMA(g_0, g_1, ...g_t) \approx \mathbb{E}(g_t), \ \ m_{t,x} \approx \mathbb{E}g_{t,x} = 1, \ \ m_{t,y} \approx \mathbb{E}g_{t,y} = 0 \qquad (2)$$

$$v_t = EMA(g_0^2, g_1^2, ...g_t^2) \approx \mathbb{E}(g_t^2), \ \ v_{t,x} \approx \mathbb{E}g_{t,x}^2 = 1, \ \ v_{t,y} \approx \mathbb{E}g_{t,y}^2 = 1. \qquad (3)$$

90



| | Step | 1 | 2 | 3 | 4 | 5 |
|---|---|---|---|---|---|---|
| | $g_x$ | 1 | 1 | 1 | 1 | 1 |
| | $g_y$ | -1 | 1 | -1 | 1 | -1 |
| Adam | $v_x$ | 1 | 1 | 1 | 1 | 1 |
| | $v_y$ | 1 | 1 | 1 | 1 | 1 |
| AdaBelief | $s_x$ | 0 | 0 | 0 | 0 | 0 |
| | $s_y$ | 1 | 1 | 1 | 1 | 1 |

Figure 2: *Left:* Consider $f(x, y) = |x| + |y|$. Blue vectors represent the gradient, and the cross represents the
optimal point. The optimizer oscillates in the $y$ direction, and keeps moving forward in the $x$ direction. *Right:*
Optimization process for the example on the left. Note that denominator $\sqrt{v_{t,x}} = \sqrt{v_{t,y}}$ for Adam, hence the
same stepsize in $x$ and $y$ direction; while $\sqrt{s_{t,x}} < \sqrt{s_{t,y}}$, hence AdaBelief takes a large step in the $x$ direction,
91   and a small step in the $y$ direction.

92   In practice, the bias correction step will further reduce the error between the EMA and its expectation
93   if $g_t$ is a stationary process [8]. Note that:

$$s_t = EMA\big((g_0 - m_0)^2, ...(g_t - m_t)^2\big) \approx \mathbb{E}\big[(g_t - \mathbb{E}g_t)^2\big] = \mathbf{Var}g_t, \ \ s_{t,x} \approx 0, \ \ s_{t,y} \approx 1 \quad (4)$$

94   An example of the analysis above is summarized in Fig. 2. From Eq. 3 and Eq. 4, note that in Adam,
95   $v_x = v_y$; this is because the update of $v_t$ only uses the amplitude of $g_t$ and ignores its sign, hence
96   the stepsize for the $x$ and $y$ direction is the same $1/\sqrt{v_{t,x}} = 1/\sqrt{v_{t,y}}$. AdaBelief considers both the
97   magnitude and sign of $g_t$, and $1/\sqrt{s_{t,x}} \gg 1/\sqrt{s_{t,y}}$, hence takes a large step in the $x$ direction and a
98   small step in the $y$ direction, which matches the behaviour of an ideal optimizer.

99   **Update direction in Adam is close to "sign descent" in low-variance case**   In this section, we
100  demonstrate that when the gradient has low variance, the update direction in Adam is close to "sign
101  descent", hence deviates from the gradient. This is also mentioned in [15].

102  Under the following assumptions: (1) assume $g_t$ is drawn from a stationary distribution, hence after
103  bias correction, $\mathbb{E}v_t = (\mathbb{E}g_t)^2 + \mathbf{Var}g_t$. (2) low-noise assumption, assume $(\mathbb{E}g_t)^2 \gg \mathbf{Var}g_t$, hence
104  we have $\mathbb{E}g_t/\sqrt{\mathbb{E}v_t} \approx \mathbb{E}g_t/\sqrt{(\mathbb{E}g_t)^2} = sign(\mathbb{E}g_t)$. (3) low-bias assumption, assume $\beta_1^t$ ($\beta_1$ to the
105  power of $t$) is small, hence $m_t$ as an estimator of $\mathbb{E}g_t$ has a small bias $\beta_1^t \mathbb{E}g_t$. Then

$$\Delta\theta_t^{Adam} = -\alpha \frac{m_t}{\sqrt{v_t}+\epsilon} \approx -\alpha \frac{\mathbb{E}g_t}{\sqrt{(\mathbb{E}g_t)^2+\mathbf{Var}g_t}+\epsilon} \approx -\alpha \frac{\mathbb{E}g_t}{||\mathbb{E}g_t||} = -\alpha \, sign(\mathbb{E}g_t) \qquad (5)$$

106  In this case, Adam behaves like a "sign descent"; in 2D cases the update is $\pm 45°$ to the axis, hence
107  deviates from the true gradient direction. The "sign update" effect might cause the generalization gap

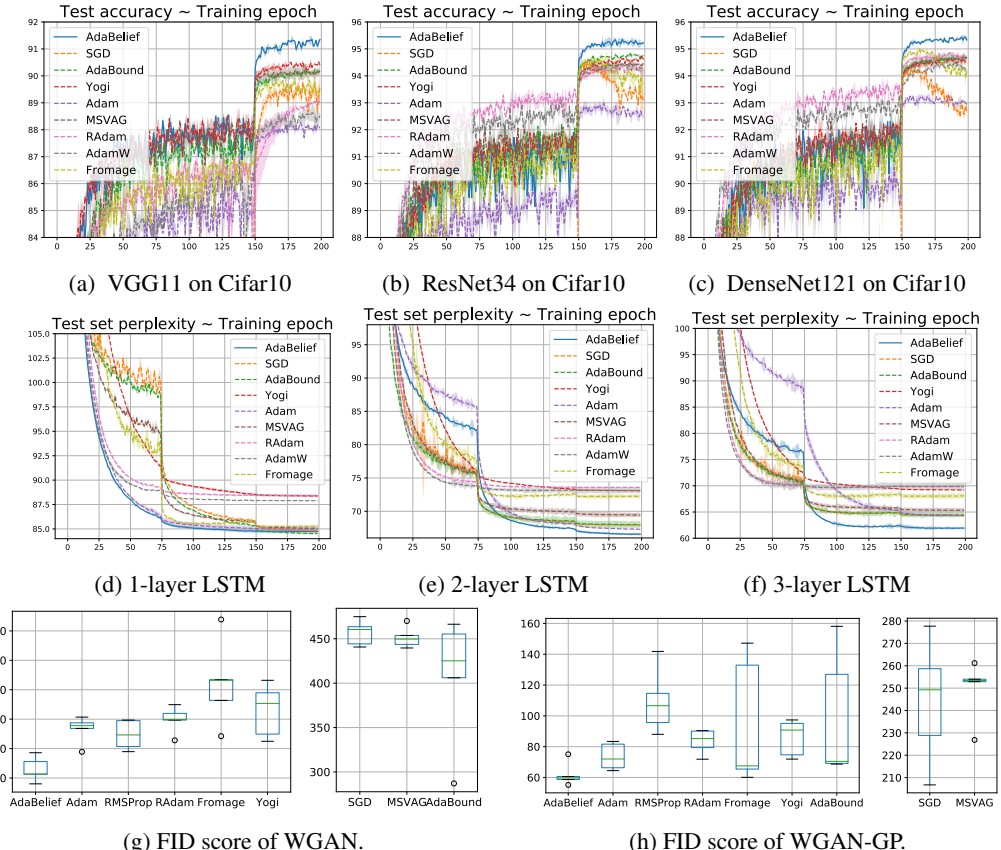

(a) VGG11 on Cifar10          (b) ResNet34 on Cifar10          (c) DenseNet121 on Cifar10

(d) 1-layer LSTM          (e) 2-layer LSTM          (f) 3-layer LSTM

(g) FID score of WGAN.          (h) FID score of WGAN-GP.

Figure 3: Top row: accuracy on Cifar10, higher is better. Middle row: perplexity on Pen-TreeBank dataset, *lower* is better. Bottom row: FID score of WGAN (GP) on Cifar10, *lower* is better.

Table 1: Top-1 accuracy of ResNet18 on ImageNet. † is reported in [22], ‡ is reported in [16]

| AdaBelief | SGD | AdaBound | Yogi | Adam | MSVAG | RAdam | AdamW |
|---|---|---|---|---|---|---|---|
| **70.08** | 70.23† | 68.13† | 68.23† | 63.79† (66.54‡) | 65.99 | 67.62‡ | 67.93† |

between adaptive methods and SGD (e.g. on ImageNet) [21, 9]. For AdaBelief, when the variance of $g_t$ is the same for all coordinates, the update direction matches the gradient direction; when the variance is not uniform, AdaBelief takes a small (large) step when the variance is large (small).

# 3   Experiments

We performed extensive comparisons with other optimizers, including SGD [3], AdaBound [12], Yogi [14], Adam [8], MSVAG [15], RAdam [16], Fromage [17] and AdamW [18]. Videos for toy examples are available[1]. The experiments include: (a) image classification on Cifar dataset [23] with VGG [24], ResNet [25] and DenseNet [26], and image recognition with ResNet on ImageNet [27]; (b) language modeling with LSTM [28] on Penn TreeBank dataset [29]; (c) wasserstein-GAN (WGAN) [30] on Cifar10 dataset. We emphasize (c) because prior work focuses on convergence and accuracy, yet neglects training stability. Results are summarized in Fig 3, and AdaBelief consistently outperforms other methods.

# 4   Conclusion

We propose the AdaBelief optimizer, which adaptively scales the stepsize by the difference between predicted gradient and observed gradient. To our knowledge, AdaBelief is the first optimizer to achieve three goals simultaneously: fast convergence as in adaptive methods, good generalization as in SGD, and training stability in complex settings such as GANs.

---

[1]`https://www.youtube.com/playlist?list=PL7KkG3n9bER6YmMLrKJ5wocjlvP7aWoOu`

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

# AdaBelief Optimizer: Adapting Stepsizes by the Belief in Observed Gradients

## Abstract

Most popular optimizers for deep learning can be broadly categorized as adaptive methods (e.g. Adam) and accelerated schemes (e.g. stochastic gradient descent (SGD) with momentum). For many models such as convolutional neural networks (CNNs), adaptive methods typically converge faster but generalize worse compared to SGD; for complex settings such as generative adversarial networks (GANs), adaptive methods are typically the default because of their stability. We propose AdaBelief to simultaneously achieve three goals: fast convergence as in adaptive methods, good generalization as in SGD, and training stability. The intuition for AdaBelief is to adapt the stepsize according to the "belief" in the current gradient direction. Viewing the exponential moving average (EMA) of the noisy gradient as the prediction of the gradient at the next time step, if the observed gradient greatly deviates from the prediction, we distrust the current observation and take a small step; if the observed gradient is close to the prediction, we trust it and take a large step. We validate AdaBelief in extensive experiments, showing that it outperforms other methods with fast convergence and high accuracy on image classification and language modeling. Specifically, on ImageNet, AdaBelief achieves comparable accuracy to SGD. Furthermore, in the training of a GAN on Cifar10, AdaBelief demonstrates high stability and improves the quality of generated samples compared to a well-tuned Adam optimizer.

## 1  Introduction

Modern neural networks are typically trained with first-order gradient methods, which can be broadly categorized into two branches: the accelerated stochastic gradient descent (SGD) family [1], such as Nesterov accelerated gradient (NAG) [2], SGD with momentum [3] and heavy-ball method (HB) [4]; and the adaptive learning rate methods, such as Adagrad [5], AdaDelta [6], RMSProp [7] and Adam [8]. SGD methods use a global learning rate for all parameters, while adaptive methods compute an individual learning rate for each parameter.

Compared to the SGD family, adaptive methods typically converge fast in the early training phases, but have poor generalization performance [9, 10]. Recent progress tries to combine the benefits of both, such as switching from Adam to SGD either with a hard schedule as in SWATS [11], or with a smooth transition as in AdaBound [12]. Other modifications of Adam are also proposed: AMSGrad [13] fixes the error in convergence analysis of Adam, Yogi [14] considers the effect of minibatch size, MSVAG [15] dissects Adam as sign update and magnitude scaling, RAdam [16] rectifies the variance of learning rate, Fromage [17] controls the distance in the function space, and AdamW [18] decouples weight decay from gradient descent. Although these modifications achieve better accuracy compared to Adam, their generalization performance is typically worse than SGD on large-scale datasets such as ImageNet [19]; furthermore, compared with Adam, many optimizers are empirically unstable when training generative adversarial networks (GAN) [20].

To solve the problems above, we propose "AdaBelief", which can be easily modified from Adam. Denote the observed gradient at step $t$ as $g_t$ and its exponential moving average (EMA) as $m_t$. Denote the EMA of $g_t^2$ and $(g_t - m_t)^2$ as $v_t$ and $s_t$, respectively. $m_t$ is divided by $\sqrt{v_t}$ in Adam, while it is divided by $\sqrt{s_t}$ in AdaBelief. Intuitively, $\frac{1}{\sqrt{s_t}}$ is the "belief" in the observation: viewing $m_t$ as the prediction of the gradient, if $g_t$ deviates much from $m_t$, we have weak belief in $g_t$, and take a small step; if $g_t$ is close to the prediction $m_t$, we have a strong belief in $g_t$, and take a large step. We validate the performance of AdaBelief with extensive experiments. Our contributions can be summarized as:

- We propose AdaBelief, which can be easily modified from Adam without extra parameters. AdaBelief has three properties: (1) fast convergence as in adaptive gradient methods, (2) good generalization as in the SGD family, and (3) training stability in complex settings such as GAN.
- We theoretically analyze the convergence property of AdaBelief in both convex optimization and non-convex stochastic optimization.
- We validate the performance of AdaBelief with extensive experiments: AdaBelief achieves fast convergence as Adam and good generalization as SGD in image classification tasks on CIFAR and ImageNet; AdaBelief outperforms other methods in language modeling; in the training of a W-GAN [30], compared to a well-tuned Adam optimizer, AdaBelief significantly improves the quality of generated images, while several recent adaptive optimizers fail the training.

## 2 Methods

### 2.1 Details of AdaBelief Optimizer

**Notations**    By the convention in [8], we use the following notations:

- $f(\theta) \in \mathbb{R}, \theta \in \mathbb{R}^d$: $f$ is the loss function to minimize, $\theta$ is the parameter in $\mathbb{R}^d$
- $\prod_{\mathcal{F},M}(y) = \mathrm{argmin}_{x \in \mathcal{F}} ||M^{1/2}(x-y)||$: projection of $y$ onto a convex feasible set $\mathcal{F}$
- $g_t$: the gradient and step $t$
- $m_t$: exponential moving average (EMA) of $g_t$
- $v_t, s_t$: $v_t$ is the EMA of $g_t^2$, $s_t$ is the EMA of $(g_t - m_t)^2$
- $\alpha, \epsilon$: $\alpha$ is the learning rate, default is $10^{-3}$; $\epsilon$ is a small number, typically set as $10^{-8}$
- $\beta_1, \beta_2$: smoothing parameters, typical values are $\beta_1 = 0.9, \beta_2 = 0.999$
- $\beta_{1t}, \beta_{2t}$ are the momentum for $m_t$ and $v_t$ respectively at step $t$, and typically set as constant (e.g. $\beta_{1t} = \beta_1, \beta_{2t} = \beta_2, \forall t \in \{1, 2, ...T\}$

| **Algorithm 1:** Adam Optimizer | **Algorithm 2:** AdaBelief Optimizer |
|---|---|
| **Initialize** $\theta_0, m_0 \leftarrow 0, v_0 \leftarrow 0, t \leftarrow 0$ | **Initialize** $\theta_0, m_0 \leftarrow 0, s_0 \leftarrow 0, t \leftarrow 0$ |
| **While** $\theta_t$ not converged | **While** $\theta_t$ not converged |
| $\quad t \leftarrow t+1$ | $\quad t \leftarrow t+1$ |
| $\quad g_t \leftarrow \nabla_\theta f_t(\theta_{t-1})$ | $\quad g_t \leftarrow \nabla_\theta f_t(\theta_{t-1})$ |
| $\quad m_t \leftarrow \beta_1 m_{t-1} + (1-\beta_1)g_t$ | $\quad m_t \leftarrow \beta_1 m_{t-1} + (1-\beta_1)g_t$ |
| $\quad v_t \leftarrow \beta_2 v_{t-1} + (1-\beta_2)g_t^2$ | $\quad s_t \leftarrow \beta_2 s_{t-1} + (1-\beta_2)(g_t - m_t)^2$ |
| $\quad$ **Update** | $\quad$ **Update** |
| $\quad\quad \theta_t \leftarrow \prod_{\mathcal{F},\sqrt{v_t}}\left(\theta_{t-1} - \frac{\alpha m_t}{\sqrt{v_t}+\epsilon}\right)$ | $\quad\quad \theta_t \leftarrow \prod_{\mathcal{F},\sqrt{s_t}}\left(\theta_{t-1} - \frac{\alpha m_t}{\sqrt{s_t}+\epsilon}\right)$ |

**Comparison with Adam**    Adam and AdaBelief are summarized in Algo.1 and Algo.2, where all operations are element-wise, with differences marked in blue. Note that no extra parameters are introduced in AdaBelief. For simplicity, we omit the bias correction step. A detailed version of AdaBelief is in Appendix A. Specifically, in Adam, the update direction is $m_t/\sqrt{v_t}$, where $v_t$ is the EMA of $g_t^2$; in AdaBelief, the update direction is $m_t/\sqrt{s_t}$, where $s_t$ is the EMA of $(g_t - m_t)^2$. Intuitively, viewing $m_t$ as the prediction of $g_t$, AdaBelief takes a large step when observation $g_t$ is close to prediction $m_t$, and a small step when the observation greatly deviates from the prediction.

### 2.2 Intuitive explanation for benefits of AdaBelief

**AdaBelief uses curvature information**    Update formulas for SGD, Adam and AdaBelief are:

$$\Delta\theta_t^{SGD} = -\alpha m_t, \ \ \Delta\theta_t^{Adam} = -\alpha m_t/\sqrt{v_t},$$
$$\Delta\theta_t^{AdaBelief} = -\alpha m_t/\sqrt{s_t} \tag{1}$$

Note that we name $\alpha$ as the "learning rate" and $|\Delta\theta_t^i|$ as the "stepsize" for the $i$th parameter. With a 1D example in Fig. 1, we demonstrate that AdaBelief uses the curvature of loss functions to improve training as summarized in Table 1, with a detailed description below:

(1) In region ① in Fig. 1, the loss function is flat, hence the gradient is close to 0. In this case, an ideal optimizer should take a large stepsize. The stepsize of SGD is proportional to the EMA of the

Table 1: Comparison of optimizers in various cases in Fig. 1. "S" and "L" represent "small" and "large" stepsize, respectively. $|\Delta\theta_t|_{ideal}$ is the stepsize of an ideal optimizer. Note that only AdaBelief matches the behaviour of an ideal optimizer in all three cases.

| | Case 1 | | | Case 2 | | | Case 3 | | |
|---|---|---|---|---|---|---|---|---|---|
| $|g_t|, v_t$ | S | | | L | | | L | | |
| $|g_t - g_{t-1}|, s_t$ | S | | | L | | | S | | |
| $|\Delta\theta_t|_{ideal}$ | L | | | S | | | L | | |
| $|\Delta\theta_t|$ | SGD | Adam | AdaBelief | SGD | Adam | AdaBelief | SGD | Adam | AdaBelief |
| | S | L | L | L | S | S | L | S | L |

gradient, hence is small in this case; while both Adam and AdaBelief take a large stepsize, because the denominator ($\sqrt{v_t}$ and $\sqrt{s_t}$) is a small value.

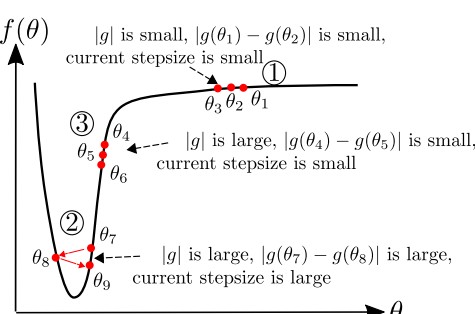

Figure 1: An ideal optimizer considers curvature of the loss function, instead of taking a large (small) step where the gradient is large (small) [31].

(2) In region ②, the algorithm oscillates in a "steep and narrow" valley, hence both $|g_t|$ and $|g_t - g_{t-1}|$ is large. An ideal optimizer should decrease its stepsize, while SGD takes a large step (proportional to $m_t$). Adam and AdaBelief take a small step because the denominator ($\sqrt{s_t}$ and $\sqrt{v_t}$) is large.

(3) In region ③, we demonstrate AdaBelief's advantage over Adam in the "large gradient, small curvature" case. In this case, $|g_t|$ and $v_t$ are large, but $|g_t - g_{t-1}|$ and $s_t$ are small; this could happen because of a small learning rate $\alpha$. In this case, an ideal optimizer should increase its stepsize. SGD uses a large stepsize ($\sim \alpha|g_t|$); in Adam, the denominator $\sqrt{v_t}$ is large, hence the stepsize is small; in AdaBelief, denominator $\sqrt{s_t}$ is small, hence the stepsize is large as in an ideal optimizer.

To sum up, AdaBelief scales the update direction by the change in gradient, which is related to the Hessian. Therefore, AdaBelief considers curvature information and performs better than Adam.

**AdaBelief considers the sign of gradient in denominator** We show the advantages of AdaBelief with a 2D example in this section, which gives us more intuition for high dimensional cases. In Fig. 2, we consider the loss function: $f(x, y) = |x| + |y|$. Note that in this simple problem, the gradient in each axis can only take $\{1, -1\}$. Suppose the start point is near the $x-$axis, e.g. $y_0 \approx 0, x_0 \ll 0$. Optimizers will oscillate in the $y$ direction, and keep increasing in the $x$ direction. Suppose the algorithm runs for a long time ($t$ is large), so the bias of EMA ($\beta_1^t \mathbb{E}g_t$) is small:

$$m_t = EMA(g_0, g_1, ...g_t) \approx \mathbb{E}(g_t), \ \ m_{t,x} \approx \mathbb{E}g_{t,x} = 1, \ \ m_{t,y} \approx \mathbb{E}g_{t,y} = 0 \qquad (2)$$

$$v_t = EMA(g_0^2, g_1^2, ...g_t^2) \approx \mathbb{E}(g_t^2), \ \ v_{t,x} \approx \mathbb{E}g_{t,x}^2 = 1, \ \ v_{t,y} \approx \mathbb{E}g_{t,y}^2 = 1. \qquad (3)$$

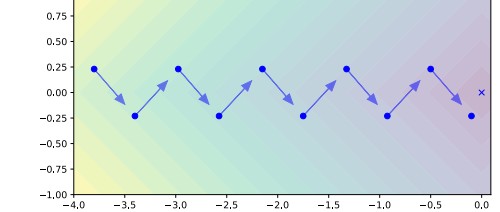

| | Step | 1 | 2 | 3 | 4 | 5 |
|---|---|---|---|---|---|---|
| | $g_x$ | 1 | 1 | 1 | 1 | 1 |
| | $g_y$ | -1 | 1 | -1 | 1 | -1 |
| Adam | $v_x$ | 1 | 1 | 1 | 1 | 1 |
| | $v_y$ | 1 | 1 | 1 | 1 | 1 |
| AdaBelief | $s_x$ | 0 | 0 | 0 | 0 | 0 |
| | $s_y$ | 1 | 1 | 1 | 1 | 1 |

Figure 2: *Left:* Consider $f(x, y) = |x| + |y|$. Blue vectors represent the gradient, and the cross represents the optimal point. The optimizer oscillates in the $y$ direction, and keeps moving forward in the $x$ direction. *Right:* Optimization process for the example on the left. Note that denominator $\sqrt{v_{t,x}} = \sqrt{v_{t,y}}$ for Adam, hence the same stepsize in $x$ and $y$ direction; while $\sqrt{s_{t,x}} < \sqrt{s_{t,y}}$, hence AdaBelief takes a large step in the $x$ direction, and a small step in the $y$ direction.

In practice, the bias correction step will further reduce the error between the EMA and its expectation if $g_t$ is a stationary process [8]. Note that:

$$s_t = EMA\big((g_0 - m_0)^2, ...(g_t - m_t)^2\big) \approx \mathbb{E}\big[(g_t - \mathbb{E}g_t)^2\big] = \mathbf{Var}g_t, \ \ s_{t,x} \approx 0, \ \ s_{t,y} \approx 1 \quad (4)$$

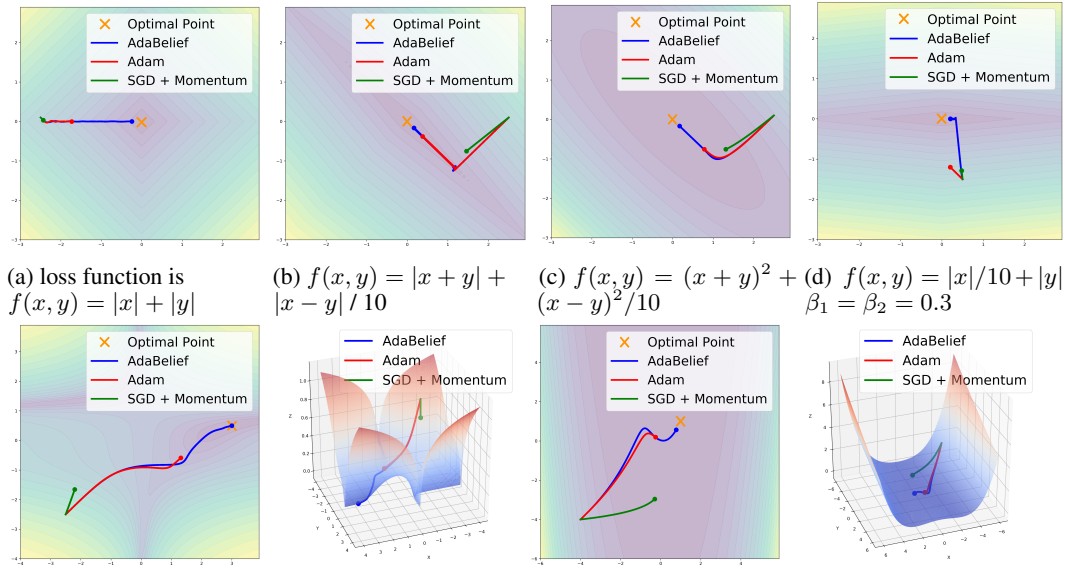

(a) loss function is $f(x, y) = |x| + |y|$

(b) $f(x, y) = |x + y| + |x - y| / 10$

(c) $f(x, y) = (x + y)^2 + (x - y)^2/10$

(d) $f(x, y) = |x|/10 + |y|$ $\beta_1 = \beta_2 = 0.3$

(e) Trajectory for Beale function in 2D.

(f) Trajectory for Beale function in 3D.

(g) Trajectory for Rosenbrock function in 2D.

(h) Trajectory for Rosenbrock function in 3D.

Figure 3: Trajectories of SGD, Adam and AdaBelief. AdaBelief reaches optimal point (marked as orange cross in 2D plots) the fastest in all cases. We refer readers to *video examples*.

An example of the analysis above is summarized in Fig. 2. From Eq. 3 and Eq. 4, note that in Adam, $v_x = v_y$; this is because the update of $v_t$ only uses the amplitude of $g_t$ and ignores its sign, hence the stepsize for the $x$ and $y$ direction is the same $1/\sqrt{v_{t,x}} = 1/\sqrt{v_{t,y}}$. AdaBelief considers both the magnitude and sign of $g_t$, and $1/\sqrt{s_{t,x}} \gg 1/\sqrt{s_{t,y}}$, hence takes a large step in the $x$ direction and a small step in the $y$ direction, which matches the behaviour of an ideal optimizer.

**Update direction in Adam is close to "sign descent" in low-variance case**   In this section, we demonstrate that when the gradient has low variance, the update direction in Adam is close to "sign descent", hence deviates from the gradient. This is also mentioned in [15].

Under the following assumptions: (1) assume $g_t$ is drawn from a stationary distribution, hence after bias correction, $\mathbb{E}v_t = (\mathbb{E}g_t)^2 + \mathbf{Var}g_t$. (2) low-noise assumption, assume $(\mathbb{E}g_t)^2 \gg \mathbf{Var}g_t$, hence we have $\mathbb{E}g_t/\sqrt{\mathbb{E}v_t} \approx \mathbb{E}g_t/\sqrt{(\mathbb{E}g_t)^2} = sign(\mathbb{E}g_t)$. (3) low-bias assumption, assume $\beta_1^t$ ($\beta_1$ to the power of $t$) is small, hence $m_t$ as an estimator of $\mathbb{E}g_t$ has a small bias $\beta_1^t\mathbb{E}g_t$. Then

$$\Delta\theta_t^{Adam} = -\alpha\frac{m_t}{\sqrt{v_t}+\epsilon} \approx -\alpha\frac{\mathbb{E}g_t}{\sqrt{(\mathbb{E}g_t)^2+\mathbf{Var}g_t}+\epsilon} \approx -\alpha\frac{\mathbb{E}g_t}{||\mathbb{E}g_t||} = -\alpha\,\text{sign}(\mathbb{E}g_t) \tag{5}$$

In this case, Adam behaves like a "sign descent"; in 2D cases the update is $\pm45°$ to the axis, hence deviates from the true gradient direction. The "sign update" effect might cause the generalization gap between adaptive methods and SGD (e.g. on ImageNet) [21, 9]. For AdaBelief, when the variance of $g_t$ is the same for all coordinates, the update direction matches the gradient direction; when the variance is not uniform, AdaBelief takes a small (large) step when the variance is large (small).

**Numerical experiments**   In this section, we validate intuitions in Sec. 2.2. Examples are shown in Fig. 3, and we refer readers to more *video examples*[2] for better visualization. In all examples, compared with SGD with momentum and Adam, AdaBelief reaches the optimal point at the fastest speed. Learning rate is $\alpha = 10^{-3}$ for all optimizers. For all examples except Fig. 3(d), we set the parameters of AdaBelief to be the same as the default in Adam [8], $\beta_1 = 0.9, \beta_2 = 0.999, \epsilon = 10^{-8}$, and set momentum as 0.9 for SGD. For Fig. 3(d), to match the assumption in Sec. 2.2, we set $\beta_1 = \beta_2 = 0.3$ for both Adam and AdaBelief, and set momentum as 0.3 for SGD.

---

[2]`https://www.youtube.com/playlist?list=PL7KkG3n9bER6YmMLrKJ5wocjlvP7aWoOu`

(a) Consider the loss function $f(x, y) = |x| + |y|$ and a starting point near the $x$ axis. This setting corresponds to Fig. 2. Under the same setting, AdaBelief takes a large step in the $x$ direction, and a small step in the $y$ direction, validating our analysis. More examples such as $f(x, y) = |x|/10 + |y|$ are in the supplementary videos.

(b) For an inseparable $L_1$ loss, AdaBelief outperforms other methods under the same setting.

(c) For an inseparable $L_2$ loss, AdaBelief outperforms other methods under the same setting.

(d) We set $\beta_1 = \beta_2 = 0.3$ for Adam and AdaBelief, and set momentum as 0.3 in SGD. This corresponds to settings of Eq. 5. For the loss $f(x, y) = |x|/10 + |y|$, $g_t$ is a constant for a large region, hence $||\mathbb{E}g_t|| \gg \mathbf{Var}g_t$. As mentioned in [8], $\mathbb{E}m_t = (1 - \beta^t)\mathbb{E}g_t$, hence a smaller $\beta$ decreases $||m_t - \mathbb{E}g_t||$ faster to 0. Adam behaves like a sign descent ($45°$ to the axis), while AdaBelief and SGD update in the direction of the gradient.

(e)-(f) Optimization trajectory under default setting for the Beale [32] function in 2D and 3D.

(g)-(h) Optimization trajectory under default setting for the Rosenbrock [33] function.

**Above cases occur frequently in deep learning** Although the above cases are simple, they give hints to local behavior of optimizers in deep learning, and we expect them to occur frequently in deep learning. Hence, we expect AdaBelief to outperform Adam in *general cases*. Other works in the literature [13, 12] claim advantages over Adam, but are typically substantiated with *carefully-constructed examples*. Note that most deep networks use ReLU activation [34], which behaves like an absolute value function as in Fig. 3(a); considering the interaction between neurons, most networks behave like case Fig. 3(b), and typically are ill-conditioned (the weight of some parameters are far larger than others) as in the figure. Considering a smooth loss function such as cross entropy or a smooth activation, this case is similar to Fig. 3(c). The case with Fig. 3(d) requires $|m_t| \approx |\mathbb{E}g_t| \gg \mathbf{Var}g_t$, and this typically occurs at the late stages of training, where the learning rate $\alpha$ is decayed to a small value, and the network reaches a stable region.

## 2.3 Convergence analysis in convex and non-convex optimization

Similar to [13, 12, 35], for simplicity, we omit the de-biasing step (analysis applicable to de-biased version). Proof for convergence in convex and non-convex cases is in the appendix.

**Optimization problem** For deterministic problems, the problem to be optimized is $\min_{\theta \in \mathcal{F}} f(\theta)$; for online optimization, the problem is $\min_{\theta \in \mathcal{F}} \sum_{t=1}^{T} f_t(\theta)$, where $f_t$ can be interpreted as loss of the model with the chosen parameters in the $t$-th step.

**Theorem 2.1.** *(Convergence in convex optimization) Let $\{\theta_t\}$ and $\{s_t\}$ be the sequence obtained by AdaBelief, let $0 \leq \beta_2 < 1, \alpha_t = \frac{\alpha}{\sqrt{t}}, \beta_{11} = \beta_1, 0 \leq \beta_{1t} \leq \beta_1 < 1, s_t \leq s_{t+1}, \forall t \in [T]$. Let $\theta \in \mathcal{F}$, where $\mathcal{F} \subset \mathbb{R}^d$ is a convex feasible set with bounded diameter $D_\infty$. Assume $f(\theta)$ is a convex function and $||g_t||_\infty \leq G_\infty/2$ (hence $||g_t - m_t||_\infty \leq G_\infty$) and $s_{t,i} \geq c > 0, \forall t \in [T], \theta \in \mathcal{F}$. Denote the optimal point as $\theta^*$. For $\theta_t$ generated with AdaBelief, we have the following bound on the regret:*

$$\sum_{t=1}^{T}[f_t(\theta_t) - f_t(\theta^*)] \leq \frac{D_\infty^2 \sqrt{T}}{2\alpha(1-\beta_1)}\sum_{i=1}^{d} s_{T,i}^{1/2} + \frac{(1+\beta_1)\alpha\sqrt{1+\log T}}{2\sqrt{c}(1-\beta_1)^3}\sum_{i=1}^{d}\left\|g_{1:T,i}^2\right\|_2 + \frac{D_\infty^2}{2(1-\beta_1)}\sum_{t=1}^{T}\sum_{i=1}^{d}\frac{\beta_{1t}s_{t,i}^{1/2}}{\alpha_t}$$

**Corollary 2.1.1.** *Suppose $\beta_{1,t} = \beta_1\lambda^t, \ 0 < \lambda < 1$ in Theorem (2.1), then we have:*

$$\sum_{t=1}^{T}[f_t(\theta_t) - f_t(\theta^*)] \leq \frac{D_\infty^2\sqrt{T}}{2\alpha(1-\beta_1)}\sum_{i=1}^{d}s_{T,i}^{1/2} + \frac{(1+\beta_1)\alpha\sqrt{1+\log T}}{2\sqrt{c}(1-\beta_1)^3}\sum_{i=1}^{d}\left\|g_{1:T,i}^2\right\|_2 + \frac{D_\infty^2\beta_1 G_\infty}{2(1-\beta_1)(1-\lambda)^2\alpha}$$

For the convex case, Theorem 2.1 implies the regret of AdaBelief is upper bounded by $O(\sqrt{T})$. Conditions for Corollary 2.1.1 can be relaxed to $\beta_{1,t} = \beta_1/t$ as in [13], which still generates $O(\sqrt{T})$ regret. Similar to Theorem 4.1 in [8] and corollary 1 in [13], where the term $\sum_{i=1}^{d} v_{T,i}^{1/2}$ exists, we have $\sum_{i=1}^{d} s_{T,i}^{1/2}$. Without further assumption, $\sum_{i=1}^{d} s_{T,i}^{1/2} < dG_\infty$ since $||g_t - m_t||_\infty < G_\infty$ as assumed in Theorem 2.1, and $dG_\infty$ is constant. The literature [8, 13, 5] exerts a stronger assumption that $\sum_{i=1}^{d} \sqrt{T}v_{T,i}^{1/2} \ll dG_\infty\sqrt{T}$. Our assumption could be similar or weaker, because $\mathbb{E}s_t = \mathbf{Var}g_t \leq \mathbb{E}g_t^2 = \mathbb{E}v_t$, then we get better regret than $O(\sqrt{T})$.

**Theorem 2.2.** *(Convergence for non-convex stochastic optimization) Under the assumptions:*

- *$f$ is differentiable; $||\nabla f(x) - \nabla f(y)|| \leq L||x - y||, \ \forall x, y$; $f$ is also lower bounded.*

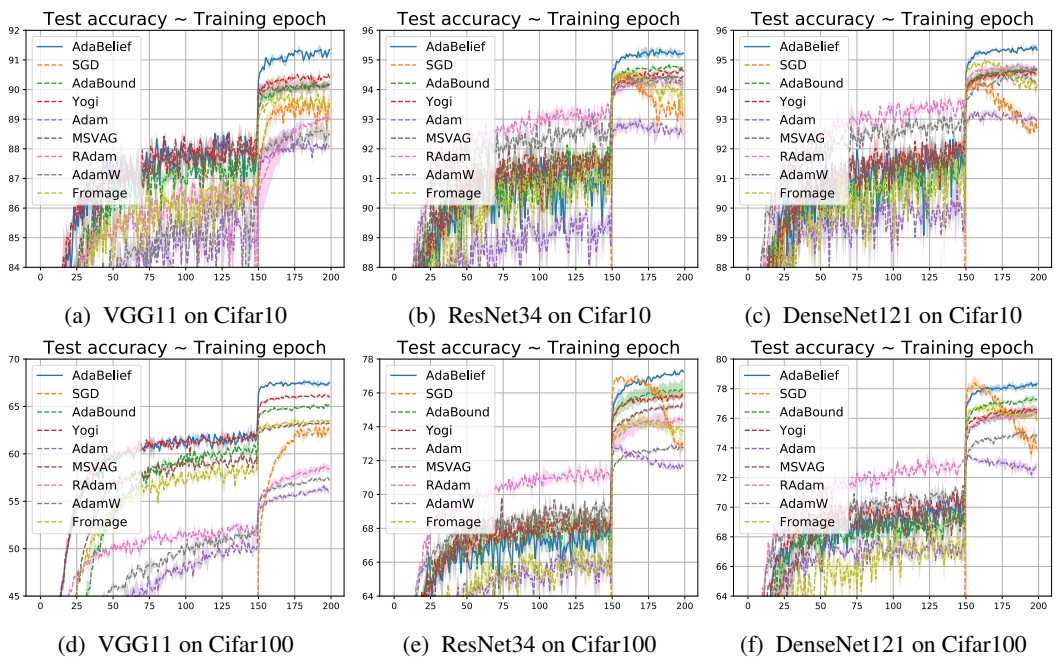

| | | |
|---|---|---|
| (a) VGG11 on Cifar10 | (b) ResNet34 on Cifar10 | (c) DenseNet121 on Cifar10 |
| (d) VGG11 on Cifar100 | (e) ResNet34 on Cifar100 | (f) DenseNet121 on Cifar100 |

Figure 4: Test accuracy ($[\mu \pm \sigma]$) on Cifar. Code modified from official implementation of AdaBound.

- *The noisy gradient is unbiased, and has independent noise, i.e. $g_t = \nabla f(\theta_t) + \zeta_t, \mathbb{E}\zeta_t = 0, \zeta_t \perp \zeta_j, \forall t, j \in \mathbb{N}, t \neq j$.*
- *At step $t$, the algorithm can access a bounded noisy gradient, and the true gradient is also bounded. i.e. $||\nabla f(\theta_t)|| \leq H, ||g_t|| \leq H, \forall t > 1$.*

*Assume $\min_{j \in [d]} (s_1)_j \geq c > 0$, noise in gradient has bounded variance, $\mathrm{Var}(g_t) = \sigma_t^2 \leq \sigma^2, \forall t \in \mathbb{N}$, then the proposed algorithm satisfies:*

$$\min_{t \in [T]} \mathbb{E} \left|\left| \nabla f(\theta_t) \right|\right|^2 \leq \frac{H}{\sqrt{T}\alpha} \left[ \frac{C_1 \alpha^2 (H^2 + \sigma^2)(1 + \log T)}{c} + C_2 \frac{d\alpha}{\sqrt{c}} + C_3 \frac{d\alpha^2}{c} + C_4 \right]$$

*as in [35], $C_1, C_2, C_3$ are constants independent of $d$ and $T$, and $C_4$ is a constant independent of $T$.*

**Corollary 2.2.1.** *If $c > C_1 H$ and assumptions for Theorem 2.2 are satisfied, we have:*

$$\frac{1}{T} \sum_{t=1}^{T} \mathbb{E} \left[ \alpha_t^2 \left|\left| \nabla f(\theta_t) \right|\right|^2 \right] \leq \frac{1}{T} \frac{1}{\frac{1}{H} - \frac{C_1}{c}} \left[ \frac{C_1 \alpha^2 \sigma^2}{c} \left( 1 + \log T \right) + C_2 \frac{d\alpha}{\sqrt{c}} + C_3 \frac{d\alpha^2}{c} + C_4 \right]$$

Theorem 2.2 implies the convergence rate for AdaBelief in the non-convex case is $O(\log T/\sqrt{T})$, which is similar to Adam-type optimizers [13, 35]. Note that regret bounds are derived in the *worst possible case*, while empirically AdaBelief outperforms Adam mainly because the cases in Sec. 2.2 occur more frequently. It is possible that the above bounds are loose; we will try to derive a tighter bound in the future.

# 3 Experiments

We performed extensive comparisons with other optimizers, including SGD [3], AdaBound [12], Yogi [14], Adam [8], MSVAG [15], RAdam [16], Fromage [17] and AdamW [18]. The experiments include: (a) image classification on Cifar dataset [23] with VGG [24], ResNet [25] and DenseNet [26], and image recognition with ResNet on ImageNet [27]; (b) language modeling with LSTM [28] on Penn TreeBank dataset [29]; (c) wasserstein-GAN (WGAN) [30] on Cifar10 dataset. We emphasize (c) because prior work focuses on convergence and accuracy, yet neglects training stability.

**Hyperparameter tuning** We performed a careful hyperparameter tuning in experiments. On image classification and language modeling we use the following:

Table 2: Top-1 accuracy of ResNet18 on ImageNet. † is reported in [22], ‡ is reported in [16]

| AdaBelief | SGD | AdaBound | Yogi | Adam | MSVAG | RAdam | AdamW |
|---|---|---|---|---|---|---|---|
| **70.08** | 70.23[†] | 68.13[†] | 68.23[†] | 63.79[†] (66.54[‡]) | 65.99 | 67.62[‡] | 67.93[†] |

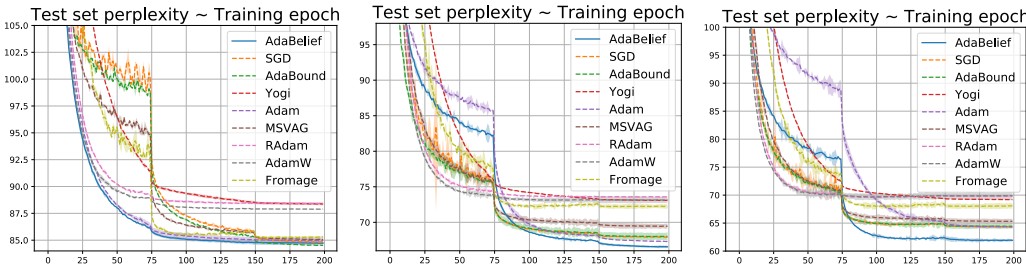

Figure 5: Left to right: perplexity ($[\mu \pm \sigma]$) on Penn Treebank for 1,2,3-layer LSTM. **Lower** is better.

474

475 • *AdaBelief:* We use the default parameters of Adam: $\beta_1 = 0.9, \beta_2 = 0.999, \epsilon = 10^{-8}, \alpha = 10^{-3}$.

476 • *SGD, Fromage:* We set the momentum as $0.9$, which is the default for many networks such as

477 ResNet [25] and DenseNet[26]. We search learning rate among $\{10.0, 1.0, 0.1, 0.01, 0.001\}$.

478 • *Adam, Yogi, RAdam, MSVAG, AdaBound:* We search for optimal $\beta_1$ among $\{0.5, 0.6, 0.7, 0.8, 0.9\}$,

479 search for $\alpha$ as in SGD, and set other parameters as their own default values in the literature.

480 • *AdamW:* We use the same parameter searching scheme as Adam. For other optimizers, we set the

481 weight decay as $5 \times 10^{-4}$; for AdamW, since the optimal weight decay is typically larger [18], we

482 search weight decay among $\{10^{-4}, 5 \times 10^{-4}, 10^{-3}, 10^{-2}\}$.

483 For the training of a GAN, we set $\beta_1 = 0.5, \epsilon = 10^{-12}$ for AdaBelief; for other methods, we search

484 for $\beta_1$ among $\{0.5, 0.6, 0.7, 0.8, 0.9\}$, and search for $\epsilon$ among $\{10^{-3}, 10^{-5}, 10^{-8}, 10^{-10}, 10^{-12}\}$.

485 We set learning rate as $2 \times 10^{-4}$ for all methods. Note that the recommended parameters for Adam

486 [36] and for RMSProp [37] are within the search range.

**CNNs on image classification** We experiment with VGG11, ResNet34 and DenseNet121 on
488 Cifar10 and Cifar100 dataset. We use the *official implementation* of AdaBound, hence achieved an
489 *exact replication* of [12]. For each optimizer, we search for the optimal hyperparameters, and report
490 the mean and standard deviation of test-set accuracy (under optimal hyperparameters) for 3 runs with
491 random initialization. As Fig. 4 shows, AdaBelief achieves fast convergence as in adaptive methods
492 such as Adam while achieving better accuracy than SGD and other methods.

493 We then train a ResNet18 on ImageNet, and report the accuracy on the validation set in Table. 2. Due
494 to the heavy computational burden, we could not perform an extensive hyperparameter search; instead,
495 we report the result of AdaBelief with the default parameters of Adam ($\beta_1 = 0.9, \beta_2 = 0.999, \epsilon = 10^{-8}$) and decoupled weight decay as in [16, 18]; for other optimizers, we report the *best result in*
497 *the literature*. AdaBelief outperforms other adaptive methods and achieves comparable accuracy to
498 SGD (70.08 v.s. 70.23), which closes the generalization gap between adaptive methods and SGD.
499 Experiments validate the fast convergence and good generalization performance of AdaBelief.

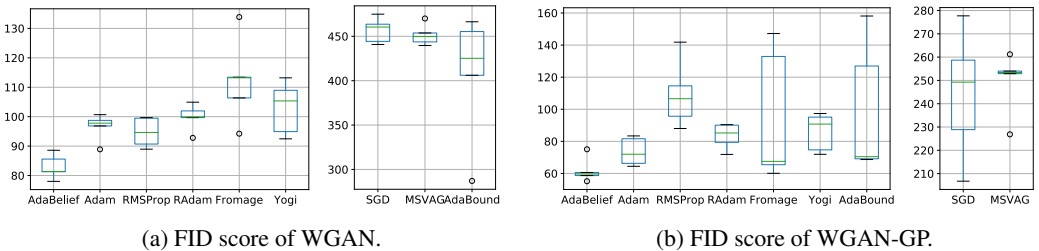

(a) FID score of WGAN.      (b) FID score of WGAN-GP.

Figure 6: FID score of WGAN and WGAN-GP on Cifar10. **Lower** is better. For each model, success
and failure optimizers are shown in the left and right respectively, with different ranges in y value.

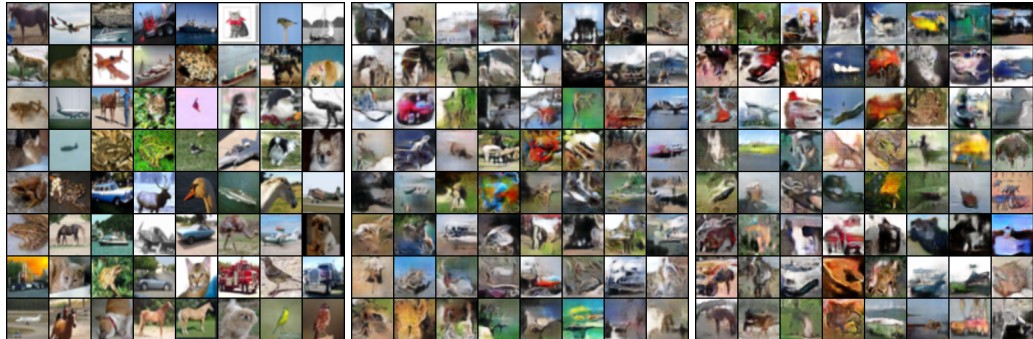

Figure 7: Left to right: real images, samples from WGAN, WGAN-GP (both trained by AdaBelief).

Table 3: Comparison of AdaBelief and Padam. Higher Acc (lower FID) is better. ‡ is from [22].

|  | AdaBelief | Padam | | | | | | |
|  |  | p=1/2 (Adam) | p=2/5 | p=1/4 | p=1/5 | p=1/8 | p=1/16 | p = 0 (SGD) |
| --- | --- | --- | --- | --- | --- | --- | --- | --- |
| ImageNet Acc | 70.08 | 63.79‡ | - | - | - | 70.07‡ | - | **70.23** ‡ |
| FID (WGAN) | **83.0± 4.1** | 96.6±4.5 | 97.5±2.8 | 426.4±49.6 | 401.5±33.2 | 328.1±37.2 | 362.6±43.9 | 469.3 ± 7.9 |
| FID (WGAN-GP) | **61.8± 7.7** | 73.5±8.7 | 87.1±6.0 | 155.1±23.8 | 167.3±27.6 | 203.6±18.9 | 228.5±25.8 | 244.3± 27.4 |

**LSTM on language modeling**  We experiment with LSTM on the Penn TreeBank dataset [29], and report the perplexity (lower is better) on the test set in Fig. 5. We report the mean and standard deviation across 3 runs. For both 2-layer and 3-layer LSTM models, AdaBelief achieves the lowest perplexity, validating its fast convergence as in adaptive methods and good accuracy. For the 1-layer model, the performance of AdaBelief is close to other optimizers.

**Generative adversarial networks**  Stability of optimizers is important in practice such as training of GANs, yet recently proposed optimizers often lack experimental validations. The training of a GAN alternates between generator and discriminator in a mini-max game, and is typically unstable [20]; SGD often generates mode collapse, and adaptive methods such as Adam and RMSProp are recommended in practice [38, 37, 39]. Therefore, training of GANs is a good test for the stability of optimizers.

We experiment with one of the most widely used models, the Wasserstein-GAN (WGAN) [30] and the improved version with gradient penalty (WGAN-GP) [37]. Using each optimizer, we train the model for 100 epochs, generate 64,000 fake images from noise, and compute the Frechet Inception Distance (FID) [40] between the fake images and real dataset (60,000 real images). FID score captures both the quality and diversity of generated images and is widely used to assess generative models (lower FID is better). For each optimizer, under its optimal hyperparameter settings, we perform 5 runs of experiments, and report the results in Fig. 6 and Fig. 7. AdaBelief significantly outperforms other optimizers, and achieves the lowest FID score.

**Remarks**  Recent research on optimizers tries to combine the fast convergence of adaptive methods with high accuracy of SGD. AdaBound [12] achieves this goal on Cifar, yet its performance on ImageNet is still inferior to SGD [22]. Padam [22] closes this generalization gap on ImageNet; writing the update as $\theta_{t+1} = \theta_t - \alpha m_t/v_t^p$, SGD sets $p = 0$, Adam sets $p = 0.5$, and Padam searches $p$ between 0 and 0.5 (outside this region Padam diverges [22, 41]). Intuitively, compared to Adam, by using a smaller $p$, Padam sacrifices the adaptivity for better generalization as in SGD; however, without good adaptivity, Padam loses training stability. As in Table 3, compared with Padam, AdaBelief achieves a much lower FID score in the training of GAN, meanwhile achieving slightly higher accuracy on ImageNet classification. Furthermore, AdaBelief has the same number of parameters as Adam, while Padam has one more parameter hence is harder to tune.

## 4   Related works

This work considers the update step in first-order methods. Other directions include Lookahead [42] which updates "fast" and "slow" weights separately, and is a wrapper that can combine with other

optimizers; variance reduction methods [43, 44, 45] which reduce the variance in gradient; and LARS [46] which uses a layer-wise learning rate scaling. AdaBelief can be combined with these methods. Other variants of Adam have been proposed (e.g. NosAdam [47], Sadam [48] and Adax [49]).

Besides first-order methods, second-order methods (e.g. Newton's method [50], Quasi-Newton method and Gauss-Newton method [51, 52, 51], L-BFGS [53], Natural-Gradient [54, 55], Conjugate-Gradient [56]) are widely used in conventional optimization. Hessian-free optimization (HFO) [57] uses second-order methods to train neural networks. Second-order methods typically use curvature information and are invariant to scaling [58] but have heavy computational burden, and hence are not widely used in deep learning.

## 5   Conclusion

We propose the AdaBelief optimizer, which adaptively scales the stepsize by the difference between predicted gradient and observed gradient. To our knowledge, AdaBelief is the first optimizer to achieve three goals simultaneously: fast convergence as in adaptive methods, good generalization as in SGD, and training stability in complex settings such as GANs. Furthermore, Adabelief has the same parameters as Adam, hence is easy to tune. We validate the benefits of AdaBelief with intuitive examples, theoretical convergence analysis in both convex and non-convex cases, and extensive experiments on real-world datasets.

## Broader Impact

Optimization is at the core of modern machine learning, and numerous efforts have been put into it. To our knowledge, AdaBelief is the first optimizer to achieve fast speed, good generalization and training stability. Adabelief can be used for the training of all models that can numerically esimate parameter gradient. hence can boost the development and application of deep learning models; yet this work mainly focuses on the theory part, and the social impact is mainly determined by each application rather than by optimizer.

# Appendix

## A. Detailed Algorithm of AdaBelief

**Notations**    By the convention in [8], we use the following notations:

- $f(\theta) \in \mathbb{R}, \theta \in \mathbb{R}^d$: $f$ is the loss function to minimize, $\theta$ is the parameter in $\mathbb{R}^d$
- $g_t$: the gradient and step $t$
- $\alpha, \epsilon$: $\alpha$ is the learning rate, default is $10^{-3}$; $\epsilon$ is a small number, typically set as $10^{-8}$
- $\beta_1, \beta_2$: smoothing parameters, typical values are $\beta_1 = 0.9, \beta_2 = 0.999$
- $m_t$: exponential moving average (EMA) of $g_t$
- $v_t, s_t$: $v_t$ is the EMA of $g_t^2$, $s_t$ is the EMA of $(g_t - m_t)^2$
- $\prod_{\mathcal{F},M}(y) = \mathrm{argmin}_{x \in \mathcal{F}} ||M^{1/2}(x - y)||$

---
**Algorithm 1: AdaBelief**

---
**Initialize** $\theta_0$
$\qquad m_0 \leftarrow 0 \ , s_0 \leftarrow 0, t \leftarrow 0$
**While** $\theta_t$ not converged
$\qquad t \leftarrow t + 1$
$\qquad g_t \leftarrow \nabla_\theta f_t(\theta_{t-1})$
$\qquad m_t \leftarrow \beta_1 m_{t-1} + (1 - \beta_1)g_t$
$\qquad s_t \leftarrow \beta_2 s_{t-1} + (1 - \beta_2)(g_t - m_t)^2$
$\qquad$ **If** $AMSGrad$
$\qquad\qquad s_t \leftarrow \max(s_t, s_{t-1})$
$\qquad$ **Bias Correction**
$\qquad\qquad \widehat{m_t} \leftarrow m_t/(1 - \beta_1^t), \qquad \widehat{s_t} \leftarrow (s_t + \epsilon)/(1 - \beta_2^t)$
$\qquad$ **Update**
$\qquad\qquad \theta_t \leftarrow \prod_{\mathcal{F},\sqrt{s_t}} \left( \theta_{t-1} - \widehat{m_t} \frac{\alpha}{\sqrt{\widehat{s_t}} + \epsilon} \right)$

---

## B. Convergence analysis in convex online learning case (Theorem 2.1 in main paper)

For the ease of notation, we absorb $\epsilon$ into $s_t$. Equivalently, $s_t \geq c > 0, \forall t \in [T]$. For simplicity, we omit the debiasing step in theoretical analysis as in [13]. Our analysis can be applied to the de-biased version as well.

**Lemma .1.** [59] For any $Q \in S_+^d$ and convex feasible set $\mathcal{F} \subset \mathbb{R}^d$, suppose $u_1 = \min_{x \in \mathcal{F}} \left|\left| Q^{1/2}(x - z_1) \right|\right|$ and $u_2 = \min_{x \in \mathcal{F}} \left|\left| Q^{1/2}(x - z_2) \right|\right|$, then we have $\left|\left| Q^{1/2}(u_1 - u_2) \right|\right| \leq \left|\left| Q^{1/2}(z_1 - z_2) \right|\right|$.

**Theorem .2.** Let $\{\theta_t\}$ and $\{s_t\}$ be the sequence obtained by the proposed algorithm, let $0 \leq \beta_2 < 1, \alpha_t = \frac{\alpha}{\sqrt{t}}, \beta_{11} = \beta_1, 0 \leq \beta_{1t} \leq \beta_1 < 1, s_{t-1} \leq s_t, \forall t \in [T]$. Let $\theta \in \mathcal{F}$, where $\mathcal{F} \subset \mathbb{R}^d$ is a convex feasible set with bounded diameter $D_\infty$. Assume $f(\theta)$ is a convex function and $||g_t||_\infty \leq G_\infty/2$ (hence $||g_t - m_t||_\infty \leq G_\infty$) and $s_{t,i} \geq c > 0, \forall t \in [T], \theta \in \mathcal{F}$. Denote the optimal point as $\theta^*$. For $\theta_t$ generated with Algorithm 1, we have the following bound on the regret:

$$\sum_{t=1}^T f_t(\theta_t) - f_t(\theta^*) \leq \frac{D_\infty^2 \sqrt{T}}{2\alpha(1 - \beta_1)} \sum_{i=1}^d s_{T,i}^{1/2} + \frac{(1 + \beta_1)\alpha\sqrt{1 + \log T}}{2\sqrt{c}(1 - \beta_1)^3} \sum_{i=1}^d \left|\left| g_{1:T,i}^2 \right|\right|_2$$

$$+ \frac{D_\infty^2}{2(1 - \beta_1)} \sum_{t=1}^T \sum_{i=1}^d \frac{\beta_{1t} s_{t,i}^{1/2}}{\alpha_t}$$

 *Proof:*

$$\theta_{t+1} = \prod_{\mathcal{F}, \sqrt{s_t}} (\theta_t - \alpha_t s_t^{-1/2} m_t) = \min_{\theta \in \mathcal{F}} \left\| s_t^{1/4} [\theta - (\theta_t - \alpha_t s_t^{-1/2} m_t)] \right\|$$

581 Note that $\prod_{\mathcal{F}, \sqrt{s_t}}(\theta^*) = \theta^*$ since $\theta^* \in \mathcal{F}$. Use $\theta_i^*$ and $\theta_{t,i}$ to denote the $i$th dimension of $\theta^*$ and $\theta_t$
582 respectively. From lemma (.1), using $u_1 = \theta_{t+1}$ and $u_2 = \theta^*$, we have:

$$\left\| s_t^{1/4}(\theta_{t+1} - \theta^*) \right\|^2 \leq \left\| s_t^{1/4}(\theta_t - \alpha_t s_t^{-1/2} m_t - \theta^*) \right\|^2$$

$$= \left\| s_t^{1/4}(\theta_t - \theta^*) \right\|^2 + \alpha_t^2 \left\| s_t^{-1/4} m_t \right\|^2 - 2\alpha_t \langle m_t, \theta_t - \theta^* \rangle$$

$$= \left\| s_t^{1/4}(\theta_t - \theta^*) \right\|^2 + \alpha_t^2 \left\| s_t^{-1/4} m_t \right\|^2$$
$$- 2\alpha_t \langle \beta_{1t} m_{t-1} + (1 - \beta_{1t}) g_t, \theta_t - \theta^* \rangle \tag{1}$$

583 Note that $\beta_1 \in [0,1)$ and $\beta_2 \in [0,1)$, rearranging inequality (1), we have:

$$\langle g_t, \theta_t - \theta^* \rangle \leq \frac{1}{2\alpha_t(1 - \beta_{1t})} \left[ \left\| s_t^{1/4}(\theta_t - \theta^*) \right\|^2 - \left\| s_t^{1/4}(\theta_{t+1} - \theta^*) \right\|^2 \right]$$

$$+ \frac{\alpha_t}{2(1 - \beta_{1t})} \left\| s_t^{-1/4} m_t \right\|^2 - \frac{\beta_{1t}}{1 - \beta_{1t}} \langle m_{t-1}, \theta_t - \theta^* \rangle$$

$$\leq \frac{1}{2\alpha_t(1 - \beta_{1t})} \left[ \left\| s_t^{1/4}(\theta_t - \theta^*) \right\|^2 - \left\| s_t^{1/4}(\theta_{t+1} - \theta^*) \right\|^2 \right]$$

$$+ \frac{\alpha_t}{2(1 - \beta_{1t})} \left\| s_t^{-1/4} m_t \right\|^2$$

$$+ \frac{\beta_{1t}}{2(1 - \beta_{1t})} \alpha_t \left\| s_t^{-1/4} m_{t-1} \right\|^2 + \frac{\beta_{1t}}{2\alpha_t(1 - \beta_{1t})} \left\| s_t^{1/4}(\theta_t - \theta^*) \right\|^2$$

$$\left( \text{Cauchy-Schwartz and Young's inequality: } ab \leq \frac{a^2\epsilon}{2} + \frac{b^2}{2\epsilon}, \forall \epsilon > 0 \right) \tag{2}$$

584 By convexity of $f$, we have:

$$\sum_{t=1}^{T} f_t(\theta_t) - f_t(\theta^*) \leq \sum_{t=1}^{T} \langle g_t, \theta_t - \theta^* \rangle$$

$$\leq \sum_{t=1}^{T} \left\{ \frac{1}{2\alpha_t(1 - \beta_{1t})} \left[ \left\| s_t^{1/4}(\theta_t - \theta^*) \right\|^2 - \left\| s_t^{1/4}(\theta_{t+1} - \theta^*) \right\|^2 \right] \right.$$

$$+ \frac{1}{2(1 - \beta_{1t})} \alpha_t \left\| s_t^{-1/4} m_t \right\|^2 + \frac{\beta_{1t}}{2(1 - \beta_{1t})} \alpha_t \left\| s_t^{-1/4} m_{t-1} \right\|^2$$

$$\left. + \frac{\beta_{1t}}{2\alpha_t(1 - \beta_{1t})} \left\| s_t^{1/4}(\theta_t - \theta^*) \right\|^2 \right\}$$

$$\left( By \; formula \; (2) \right)$$

$$\leq \frac{1}{2(1 - \beta_1)} \frac{\left\| s_1^{1/4}(\theta_1 - \theta^*) \right\|^2}{\alpha_1}$$

$$+ \frac{1}{2(1 - \beta_1)} \sum_{t=2}^{T} \left[ \frac{\left\| s_t^{1/4}(\theta_t - \theta^*) \right\|^2}{\alpha_t} - \frac{\left\| s_{t-1}^{1/4}(\theta_t - \theta^*) \right\|^2}{\alpha_{t-1}} \right]$$

$$+ \sum_{t=1}^{T} \left[ \frac{1}{2(1 - \beta_1)} \alpha_t \left\| s_t^{-1/4} m_t \right\|^2 \right] + \sum_{t=2}^{T} \left[ \frac{\beta_1}{2(1 - \beta_1)} \alpha_{t-1} \left\| s_{t-1}^{-1/4} m_{t-1} \right\|^2 \right]$$

$$+ \sum_{t=1}^{T} \frac{\beta_{1t}}{2\alpha_t(1 - \beta_{1t})} \left\| s_t^{1/4}(\theta_t - \theta^*) \right\|^2$$

$$\left(0 \le s_{t-1} \le s_t, 0 \le \alpha_t \le \alpha_{t-1}, 0 \le \beta_{1t} \le \beta_1 < 1\right)$$

$$\le \frac{1}{2(1-\beta_1)} \frac{\left\|s_1^{1/4}(\theta_1 - \theta^*)\right\|^2}{\alpha_1} + \frac{1}{2(1-\beta_1)} \sum_{t=2}^{T} \left\|\theta_t - \theta^*\right\|^2 \left[\frac{s_t^{1/2}}{\alpha_t} - \frac{s_{t-1}^{1/2}}{\alpha_{t-1}}\right]$$

$$+ \frac{1+\beta_1}{2(1-\beta_1)} \sum_{t=1}^{T} \alpha_t \left\|s_t^{-1/4} m_t\right\|^2$$

$$+ \sum_{t=1}^{T} \frac{\beta_{1t}}{2\alpha_t(1-\beta_{1t})} \left\|s_t^{1/4}(\theta_t - \theta^*)\right\|^2$$

$$\le \frac{1}{2(1-\beta_1)} \frac{\left\|s_1^{1/4}(\theta_1 - \theta^*)\right\|^2}{\alpha_1} + \frac{1}{2(1-\beta_1)} \sum_{t=2}^{T} \left\|\theta_t - \theta^*\right\|^2 \left[\frac{s_t^{1/2}}{\alpha_t} - \frac{s_{t-1}^{1/2}}{\alpha_{t-1}}\right]$$

$$+ \frac{1+\beta_1}{2(1-\beta_1)} \sum_{t=1}^{T} \alpha_t \left\|s_t^{-1/4} m_t\right\|^2$$

$$+ \frac{1}{2(1-\beta_1)} \sum_{t=1}^{T} \frac{\beta_{1t}}{\alpha_t} \left\|s_t^{1/4}(\theta_t - \theta^*)\right\|^2$$

$$\left(since\ 0 \le \beta_{1t} \le \beta_1 < 1\right) \tag{3}$$

585  Now bound $\sum_{t=1}^{T} \alpha_t \|s_t^{-1/4} m_t\|^2$ in Formula (3), assuming $0 < c \le s_t, \forall t \in [T]$.

$$\sum_{t=1}^{T} \alpha_t \left\|s_t^{-1/4} m_t\right\|^2 = \sum_{t=1}^{T-1} \alpha_t \left\|s_t^{-1/4} m_t\right\|^2 + \alpha_T \left\|s_T^{-1/4} m_T\right\|^2$$

$$\le \sum_{t=1}^{T-1} \alpha_t \left\|s_t^{-1/4} m_t\right\|^2 + \frac{\alpha_T}{\sqrt{c}} \left\|m_T\right\|^2$$

$$= \sum_{t=1}^{T-1} \alpha_t \left\|s_t^{-1/4} m_t\right\|^2 + \frac{\alpha}{\sqrt{cT}} \sum_{i=1}^{d} \left(\sum_{j=1}^{T}(1-\beta_{1,j}) g_{j,i} \prod_{k=1}^{T-j} \beta_{1,T-k+1}\right)^2$$

$$\left(since\ m_T = \sum_{j=1}^{T}(1-\beta_{1,j}) g_{j,i} \prod_{k=1}^{T-j} \beta_{1,T-k+1}\right)$$

$$\le \sum_{t=1}^{T-1} \alpha_t \left\|s_t^{-1/4} m_t\right\|^2 + \frac{\alpha}{\sqrt{cT}} \sum_{i=1}^{d} \left(\sum_{j=1}^{T} g_{j,i} \prod_{k=1}^{T-j} \beta_1\right)^2$$

$$(since\ 0 < \beta_{1,j} \le \beta_1 < 1)$$

$$= \sum_{t=1}^{T-1} \alpha_t \left\|s_t^{-1/4} m_t\right\|^2 + \frac{\alpha}{\sqrt{cT}} \sum_{i=1}^{d} \left(\sum_{j=1}^{T} \beta_1^{T-j} g_{j,i}\right)^2$$

$$\le \sum_{t=1}^{T-1} \alpha_t \left\|s_t^{-1/4} m_t\right\|^2 + \frac{\alpha}{\sqrt{cT}} \sum_{i=1}^{d} \left(\sum_{j=1}^{T} \beta_1^{T-j}\right)\left(\sum_{j=1}^{T} \beta_1^{T-j} g_{j,i}^2\right)$$

$$\left(Cauchy - Schwartz, \langle u,v\rangle^2 \le \left\|u\right\|^2 \left\|v\right\|^2, u_j = \sqrt{\beta_1^{T-j}}, v_j = \sqrt{\beta_1^{T-j}} g_{j,i}\right)$$

$$= \sum_{t=1}^{T-1} \alpha_t \left\|s_t^{-1/4} m_t\right\|^2 + \frac{\alpha}{\sqrt{cT}} \sum_{i=1}^{d} \frac{1-\beta_1^T}{1-\beta_1} \sum_{j=1}^{T} \beta_1^{T-j} g_{j,i}^2$$

$$\le \sum_{t=1}^{T-1} \alpha_t \left\|s_t^{-1/4} m_t\right\|^2 + \frac{\alpha}{\sqrt{c}(1-\beta_1)} \sum_{i=1}^{d}\sum_{j=1}^{T} \beta_1^{T-j} g_{j,i}^2 \frac{1}{\sqrt{T}}$$

$$\left(since\ 1 - \beta_1^T < 1\right)$$

$$\leq \frac{\alpha}{\sqrt{c}(1-\beta_1)} \sum_{i=1}^{d} \sum_{t=1}^{T} \sum_{j=1}^{t} \beta_1^{t-j} g_{j,i}^2 \frac{1}{\sqrt{t}}$$

$$\left(Recursively\ bound\ each\ term\ in\ the\ sum\ \sum_{t=1}^{T} *\right)$$

$$= \frac{\alpha}{\sqrt{c}(1-\beta_1)} \sum_{i=1}^{d} \sum_{t=1}^{T} g_{t,i}^2 \sum_{j=t}^{T} \frac{\beta_1^{j-t}}{\sqrt{j}}$$

$$\leq \frac{\alpha}{\sqrt{c}(1-\beta_1)} \sum_{i=1}^{d} \sum_{t=1}^{T} g_{t,i}^2 \sum_{j=t}^{T} \frac{\beta_1^{j-t}}{\sqrt{t}}$$

$$\leq \frac{\alpha}{\sqrt{c}(1-\beta_1)^2} \sum_{i=1}^{d} \sum_{t=1}^{T} g_{t,i}^2 \frac{1}{\sqrt{t}}$$

$$\left(since\ \sum_{j=t}^{T} \beta_1^{j-t} = \sum_{j=0}^{T-t} \beta_1^j = \frac{1-\beta_1^{T-t+1}}{1-\beta_1} \leq \frac{1}{1-\beta_1}\right)$$

$$\leq \frac{\alpha}{\sqrt{c}(1-\beta_1)^2} \sum_{i=1}^{d} \left\|g_{1:T,i}^2\right\|_2 \sqrt{\sum_{t=1}^{T} \frac{1}{t}}$$

$$\left(Cauchy - Schwartz,\ \langle u, v \rangle \leq \left\|u\right\|\left\|v\right\|,\ u_t = g_{t,i}^2,\ v_t = \frac{1}{\sqrt{t}}\right)$$

$$\leq \frac{\alpha\sqrt{1+\log T}}{\sqrt{c}(1-\beta_1)^2} \sum_{i=1}^{d} \left\|g_{1:T,i}^2\right\|_2 \quad \left(since\ \sum_{t=1}^{T} \frac{1}{t} \leq 1 + \log T\right) \tag{4}$$

586  Apply formula (4) to (3), we have:

$$\sum_{t=1}^{T} f_t(\theta_t) - f_t(\theta^*) \leq \frac{1}{2(1-\beta_1)} \frac{\left\|s_1^{1/4}(\theta_1 - \theta^*)\right\|^2}{\alpha_1} + \frac{1}{2(1-\beta_1)} \sum_{t=2}^{T} \left\|\theta_t - \theta^*\right\|^2 \left[\frac{s_t^{1/2}}{\alpha_t} - \frac{s_{t-1}^{1/2}}{\alpha_{t-1}}\right]$$

$$+ \frac{1+\beta_1}{2(1-\beta_1)} \sum_{t=1}^{T} \alpha_t \left\|s_t^{-1/4} m_t\right\|^2$$

$$+ \frac{1}{2(1-\beta_1)} \sum_{t=1}^{T} \frac{\beta_{1t}}{\alpha_t} \left\|s_t^{1/4}(\theta_t - \theta^*)\right\|^2$$

$$\leq \frac{1}{2(1-\beta_1)} \frac{\left\|s_1^{1/4}(\theta_1 - \theta^*)\right\|^2}{\alpha_1} + \frac{1}{2(1-\beta_1)} \sum_{t=2}^{T} \left\|\theta_t - \theta^*\right\|^2 \left[\frac{s_t^{1/2}}{\alpha_t} - \frac{s_{t-1}^{1/2}}{\alpha_{t-1}}\right]$$

$$+ \frac{(1+\beta_1)\alpha\sqrt{1+\log T}}{2\sqrt{c}(1-\beta_1)^3} \sum_{i=1}^{d} \left\|g_{1:T,i}^2\right\|_2$$

$$+ \frac{1}{2(1-\beta_1)} \sum_{t=1}^{T} \frac{\beta_{1t}}{\alpha_t} \left\|s_t^{1/4}(\theta_t - \theta^*)\right\|^2$$

$$\left(By\ formula\ (4)\right)$$

$$\leq \frac{1}{2(1-\beta_1)} \sum_{i=1}^{d} \frac{s_{1,i}^{1/2} D_\infty^2}{\alpha_1} + \frac{1}{2(1-\beta_1)} \sum_{t=2}^{T} \sum_{i=1}^{d} D_\infty^2 \left[\frac{s_{t,i}^{1/2}}{\alpha_t} - \frac{s_{t-1,i}^{1/2}}{\alpha_{t-1}}\right]$$

$$+ \frac{(1 + \beta_1)\alpha\sqrt{1 + \log T}}{2\sqrt{c}(1 - \beta_1)^3} \sum_{i=1}^{d} \left\| g_{1:T,i}^2 \right\|_2$$

$$+ \frac{D_\infty^2}{2(1 - \beta_1)} \sum_{t=1}^{T} \sum_{i=1}^{d} \frac{\beta_{1t} s_{t,i}^{1/2}}{\alpha_t}$$

$$\left( \textit{since } x \in \mathcal{F}, \textit{with bounded diameter } D_\infty, \textit{and } \frac{s_{t,i}^{1/2}}{\alpha_t} \geq \frac{s_{t-1,i}^{1/2}}{\alpha_{t-1}} \textit{ by assumption.} \right)$$

$$\leq \frac{D_\infty^2 \sqrt{T}}{2\alpha(1 - \beta_1)} \sum_{i=1}^{d} s_{T,i}^{1/2} + \frac{(1 + \beta_1)\alpha\sqrt{1 + \log T}}{2\sqrt{c}(1 - \beta_1)^3} \sum_{i=1}^{d} \left\| g_{1:T,i}^2 \right\|_2$$

$$+ \frac{D_\infty^2}{2(1 - \beta_1)} \sum_{t=1}^{T} \sum_{i=1}^{d} \frac{\beta_{1t} s_{t,i}^{1/2}}{\alpha_t}$$

$$\left( \alpha_t \geq \alpha_{t+1} \textit{ and perform telescope sum} \right) \tag{5}$$

587 $\qquad\qquad\qquad\qquad\qquad\qquad\qquad\qquad\qquad\qquad\qquad\qquad\qquad\qquad\qquad\qquad\qquad$ $\square$

588 **Corollary .2.1.** *Suppose* $\beta_{1,t} = \beta_1 \lambda^t$, $0 < \lambda < 1$ *in Theorem* (.2)*, then we have:*

$$\sum_{t=1}^{T} f_t(\theta_t) - f_t(\theta^*) \leq \frac{D_\infty^2 \sqrt{T}}{2\alpha(1 - \beta_1)} \sum_{i=1}^{d} s_{T,i}^{1/2} + \frac{(1 + \beta_1)\alpha\sqrt{1 + \log T}}{2\sqrt{c}(1 - \beta_1)^3} \sum_{i=1}^{d} \left\| g_{1:T,i}^2 \right\|_2$$

$$+ \frac{D_\infty^2 \beta_1 G_\infty}{2(1 - \beta_1)(1 - \lambda)^2 \alpha} \tag{6}$$

589 *Proof:* By sum of arithmetico-geometric series, we have:

$$\sum_{t=1}^{T} \lambda^{t-1} \sqrt{t} \leq \sum_{t=1}^{T} \lambda^{t-1} t \leq \frac{1}{(1 - \lambda)^2} \tag{7}$$

590 Plugging (7) into (5), we can derive the results above. $\qquad\qquad\qquad\qquad\qquad$ $\square$

## 591 C. Convergence analysis for non-convex stochastic optimization (Theorem 2.2
## 592 in main paper)

### 593 Assumptions

594 $\quad$ • A1, $f$ is differentiable and has $L - Lipschitz$ gradient, $||\nabla f(x) - \nabla f(y)|| \leq L||x -$
595 $\quad\quad$ $y||$, $\forall x, y$. $f$ is also lower bounded.
596 $\quad$ • A2, at time $t$, the algorithm can access a bounded noisy gradient, the true gradient is also
597 $\quad\quad$ bounded. *i.e.* $||\nabla f(\theta_t)|| \leq H$, $||g_t|| \leq H$, $\forall t > 1$.
598 $\quad$ • A3, The noisy gradient is unbiased, and has independent noise. *i.e.* $g_t = \nabla f(\theta_t) + \zeta_t$, $\mathbb{E}\zeta_t =$
599 $\quad\quad$ $0, \zeta_t \perp \zeta_j$, $\forall j, t \in \mathbb{N}, t \neq j$

600 **Theorem .3.** *[35] Suppose assumptions A1-A3 are satisfied,* $\beta_{1,t}$ *is chosen such that* $0 \leq \beta_{1,t+1} \leq$
601 $\beta_{1,t} < 1, 0 < \beta_2 < 1, \forall t > 0$*. For some constant* $G$*,* $\left\| \alpha_t \frac{m_t}{\sqrt{s_t}} \right\| \leq G, \forall t$*. Then Adam-type algorithms*
602 *yield*

$$\mathbb{E}\left[ \sum_{t=1}^{T} \alpha_t \langle \nabla f(\theta_t), \nabla f(\theta_t)/\sqrt{s_t} \rangle \right] \leq$$

$$\mathbb{E}\left[ C_1 \sum_{t=1}^{T} \left\| \alpha_t g_t / \sqrt{s_t} \right\|^2 + C_2 \sum_{t=1}^{T} \left\| \frac{\alpha_t}{\sqrt{s_t}} - \frac{\alpha_{t-1}}{\sqrt{s_{t-1}}} \right\|_1 + C_3 \sum_{t=1}^{T} \left\| \frac{\alpha_t}{\sqrt{s_t}} - \frac{\alpha_{t-1}}{\sqrt{s_{t-1}}} \right\|^2 \right] + C_4 \tag{8}$$

603 *where* $C_1, C_2, C_3$ *are constants independent of* $d$ *and* $T$*,* $C_4$ *is a constant independent of* $T$*, the*
604 *expectation is taken w.r.t all randomness corresponding to* $\{g_t\}$*.*

*Furthermore, let $\gamma_t := \min_{j\in[d]} \min_{\{g_i\}_{i=1}^t} \alpha_i/(\sqrt{s_i})_j$ denote the minimum possible value of effective stepsize at time $t$ over all possible coordinate and past gradients $\{g_i\}_{i=1}^t$. The convergence rate of Adam-type algorithm is given by*

$$\min_{t\in[T]} \mathbb{E}\left[\left\|\nabla f(\theta_t)\right\|^2\right] = O\left(\frac{s_1(T)}{s_2(T)}\right) \tag{9}$$

*where $s_1(T)$ is defined through the upper bound of RHS of (8), and $\sum_{t=1}^T \gamma_t = \Omega(s_2(T))$*

**Proof:** We provide the proof from [35] in next section for completeness. □

**Theorem .4.** *Assume $\min_{j\in[d]}(s_1)_j \geq c > 0$, noise in gradient has bounded variance, $\mathrm{Var}(g_t) = \sigma_t^2 \leq \sigma^2, \forall t \in \mathbb{N}$, then the AdaBelief algorithm satisfies:*

$$\min_{t\in[T]} \mathbb{E}\left\|\nabla f(\theta_t)\right\|^2 \leq \frac{H}{\sqrt{T}\alpha}\left[\frac{C_1\alpha^2(H^2+\sigma^2)(1+\log T)}{c} + C_2\frac{d\alpha}{\sqrt{c}} + C_3\frac{d\alpha^2}{c} + C_4\right]$$

$$= \frac{1}{\sqrt{T}}(Q_1 + Q_2\log T)$$

*where*

$$Q_1 = \frac{H}{\alpha}\left[\frac{C_1\alpha^2(H^2+\sigma^2)}{c} + C_2\frac{d\alpha}{\sqrt{c}} + C_3\frac{d\alpha^2}{c} + C_4\right]$$

$$Q_2 = \frac{HC_1\alpha(H^2+\sigma^2)}{c}$$

**Proof:** We first derive an upper bound of the RHS of formula (8), then derive a lower bound of the LHS of (8).

$$\mathbb{E}\left[\sum_{t=1}^T \left\|\alpha_t g_t/\sqrt{s_t}\right\|^2\right] \leq \frac{1}{c}\mathbb{E}\left[\sum_{t=1}^T\sum_{i=1}^d (\alpha_{t,i}g_{t,i})^2\right] \quad \left(since\ 0 < c \leq s_t, \forall t \in [T]\right)$$

$$= \frac{1}{c}\sum_{i=1}^d\sum_{t=1}^T \alpha_t^2\mathbb{E}(g_{t,i})^2$$

$$= \frac{1}{c}\sum_{t=1}^T \alpha_t^2\mathbb{E}\left[\left\|\nabla f(\theta_t)\right\|^2 + \left\|\sigma_t\right\|^2\right] \tag{10}$$

$$\mathbb{E}\left[\sum_{t=1}^T \left\|\frac{\alpha_t}{\sqrt{s_t}} - \frac{\alpha_{t-1}}{\sqrt{s_{t-1}}}\right\|_1\right] = \mathbb{E}\left[\sum_{i=1}^d\sum_{t=1}^T \frac{\alpha_{t-1}}{\sqrt{s_{t-1,i}}} - \frac{\alpha_t}{\sqrt{s_{t,i}}}\right]$$

$$\left(since\ \alpha_t \leq \alpha_{t-1}, s_{t,i} \geq s_{t-1,i}\right)$$

$$= \mathbb{E}\left[\sum_{i=1}^d \frac{\alpha_1}{\sqrt{s_{1,i}}} - \frac{\alpha_T}{\sqrt{s_{T,i}}}\right]$$

$$\leq \mathbb{E}\left[\sum_{i=1}^d \frac{\alpha_1}{\sqrt{s_{1,i}}}\right]$$

$$\leq \frac{d\alpha}{\sqrt{c}} \quad \left(since\ 0 < c \leq s_t, 0 \leq \alpha_t \leq \alpha_1 = \alpha, \forall t\right) \tag{11}$$

$$\mathbb{E}\left[\sum_{t=1}^T \left\|\frac{\alpha_t}{\sqrt{s_t}} - \frac{\alpha_{t-1}}{\sqrt{s_{t-1}}}\right\|^2\right] = \mathbb{E}\left[\sum_{t=1}^T\sum_{i=1}^d \left(\frac{\alpha_t}{\sqrt{s_t}} - \frac{\alpha_{t-1}}{\sqrt{s_{t-1}}}\right)_i^2\right]$$

$$\leq \mathbb{E}\Big[\sum_{t=1}^{T}\sum_{i=1}^{d}\Big|\frac{\alpha_t}{\sqrt{s_t}} - \frac{\alpha_{t-1}}{\sqrt{s_{t-1}}}\Big|_i \frac{\alpha}{\sqrt{c}}\Big]$$

$$\Big(Since \ \Big|\frac{\alpha_t}{\sqrt{s_t}} - \frac{\alpha_{t-1}}{\sqrt{s_{t-1}}}\Big| = \frac{\alpha_{t-1}}{\sqrt{s_{t-1}}} - \frac{\alpha_t}{\sqrt{s_t}} \leq \frac{\alpha_{t-1}}{\sqrt{s_{t-1}}} \leq \frac{\alpha}{\sqrt{c}}\Big)$$

$$\leq \frac{d\alpha^2}{c} \ \Big(By \ (11)\Big) \tag{12}$$

Next we derive the lower bound of LHS of (8).

$$\mathbb{E}\Big[\sum_{t=1}^{T}\alpha_t\langle\nabla f(\theta_t), \frac{\nabla f(\theta_t)}{\sqrt{s_t}}\rangle\Big] \geq \frac{1}{H}\mathbb{E}\Big[\sum_{t=1}^{T}\alpha_t\big|\big|\nabla f(\theta_t)\big|\big|^2\Big] \geq \frac{\alpha\sqrt{T}}{H}\min_{t\in[T]}\mathbb{E}\big|\big|\nabla f(\theta_t)\big|\big|^2 \tag{13}$$

Combining (10), (11), (12) and (13) to (8), we have:

$$\frac{\alpha\sqrt{T}}{H}\min_{t\in[T]}\mathbb{E}\big|\big|\nabla f(\theta_t)\big|\big|^2 \leq \mathbb{E}\Big[\sum_{t=1}^{T}\alpha_t\langle\nabla f(\theta_t), \frac{\nabla f(\theta_t)}{\sqrt{s_t}}\rangle\Big]$$

$$\leq \mathbb{E}\Big[C_1\sum_{t=1}^{T}\big|\big|\alpha_t g_t/\sqrt{s_t}\big|\big|^2 + C_2\sum_{t=1}^{T}\big|\big|\frac{\alpha_t}{\sqrt{s_t}} - \frac{\alpha_{t-1}}{\sqrt{s_{t-1}}}\big|\big|_1 + C_3\sum_{t=1}^{T}\big|\big|\frac{\alpha_t}{\sqrt{s_t}} - \frac{\alpha_{t-1}}{\sqrt{s_{t-1}}}\big|\big|^2\Big] + C_4$$

$$\leq \frac{C_1}{c}\sum_{t=1}^{T}\mathbb{E}\Big[\alpha_t^2\big|\big|\nabla f(\theta_t)\big|\big|^2 + \alpha_t^2\big|\big|\sigma_t\big|\big|^2\Big] + C_2\frac{d\alpha}{\sqrt{c}} + C_3\frac{d\alpha^2}{c} + C_4 \tag{14}$$

$$\leq \frac{C_1}{c}\sum_{t=1}^{T}\mathbb{E}\Big[\alpha_t^2(H^2 + \sigma^2)\Big] + C_2\frac{d\alpha}{\sqrt{c}} + C_3\frac{d\alpha^2}{c} + C_4$$

$$\leq \frac{C_1\alpha^2(H^2 + \sigma^2)(1 + \log T)}{c} + C_2\frac{d\alpha}{\sqrt{c}} + C_3\frac{d\alpha^2}{c} + C_4 \tag{15}$$

$$\Big(since \ \alpha_t = \frac{\alpha}{\sqrt{t}}, \ \sum_{t=1}^{T}\frac{1}{t} \leq 1 + \log T\Big)$$

Re-arranging above inequality, we have

$$\min_{t\in[T]}\mathbb{E}\big|\big|\nabla f(\theta_t)\big|\big|^2 \leq \frac{H}{\sqrt{T}\alpha}\Big[\frac{C_1\alpha^2(H^2 + \sigma^2)(1 + \log T)}{c} + C_2\frac{d\alpha}{\sqrt{c}} + C_3\frac{d\alpha^2}{c} + C_4\Big]$$

$$= \frac{1}{\sqrt{T}}(Q_1 + Q_2\log T) \tag{16}$$

where

$$Q_1 = \frac{H}{\alpha}\Big[\frac{C_1\alpha^2(H^2 + \sigma^2)}{c} + C_2\frac{d\alpha}{\sqrt{c}} + C_3\frac{d\alpha^2}{c} + C_4\Big] \tag{17}$$

$$Q_2 = \frac{HC_1\alpha(H^2 + \sigma^2)}{c} \tag{18}$$

$\square$

**Corollary .4.1.** *If $c > C_1 H$ and assumptions for Theorem .3 are satisfied, we have:*

$$\frac{1}{T}\sum_{t=1}^{T}\mathbb{E}\Big[\alpha_t^2\big|\big|\nabla f(\theta_t)\big|\big|^2\Big] \leq \frac{1}{T}\frac{1}{\frac{1}{H} - \frac{C_1}{c}}\Big[\frac{C_1\alpha^2\sigma^2}{c}(1 + \log T) + C_2\frac{d\alpha}{\sqrt{c}} + C_3\frac{d\alpha^2}{c} + C_4\Big] \tag{19}$$

*Proof:* From (13) and (14), we have

$$\frac{1}{H}\mathbb{E}\Big[\sum_{t=1}^{T}\alpha_t\big|\big|\nabla f(\theta_t)\big|\big|^2\Big] \leq \mathbb{E}\Big[\sum_{t=1}^{T}\alpha_t\langle\nabla f(\theta_t), \frac{\nabla f(\theta_t)}{\sqrt{s_t}}\rangle\Big]$$

$$\leq \frac{C_1}{c} \sum_{t=1}^{T} \mathbb{E}\Big[\alpha_t^2 \big\|\nabla f(\theta_t)\big\|^2 + \alpha_t^2 \big\|\sigma_t\big\|^2\Big] + C_2 \frac{d\alpha}{\sqrt{c}} + C_3 \frac{d\alpha^2}{c} + C_4 \tag{20}$$

By re-arranging, we have

$$\big(\frac{1}{H} - \frac{C_1}{c}\big) \sum_{t=1}^{T} \mathbb{E}\Big[\alpha_t^2 \big\|\nabla f(\theta_t)\big\|^2\Big] \leq \frac{C_1}{c} \sum_{t=1}^{T} \mathbb{E}\Big[\alpha_t^2 \big\|\sigma_t\big\|^2\Big] + C_2 \frac{d\alpha}{\sqrt{c}} + C_3 \frac{d\alpha^2}{c} + C_4$$

$$\leq \frac{C_1 \alpha^2 \sigma^2}{c}\big(1 + \log T\big) + C_2 \frac{d\alpha}{\sqrt{c}} + C_3 \frac{d\alpha^2}{c} + C_4 \tag{21}$$

By assumption, $\frac{1}{H} - \frac{C_1}{c} > 0$, then we have

$$\sum_{t=1}^{T} \mathbb{E}\Big[\alpha_t^2 \big\|\nabla f(\theta_t)\big\|^2\Big] \leq \frac{1}{\frac{1}{H} - \frac{C_1}{c}} \left[ \frac{C_1 \alpha^2 \sigma^2}{c}\big(1 + \log T\big) + C_2 \frac{d\alpha}{\sqrt{c}} + C_3 \frac{d\alpha^2}{c} + C_4 \right] \tag{22}$$

$\square$

# D. Proof of Theorem .3

**Lemma .5.** *[35] Let $\theta_0 \triangleq \theta_1$ in the Algorithm, consider the sequence*

$$z_t = \theta_t + \frac{\beta_{1,t}}{1 - \beta_{1,t}}(\theta_t - \theta_{t-1}), \forall t \geq 2$$

*The following holds true:*

$$z_{t+1} - z_t = -\Big(\frac{\beta_{1,t+1}}{1 - \beta_{1,t+1}} - \frac{\beta_{1,t}}{1 - \beta_{1,t}}\Big)\frac{\alpha_t m_t}{\sqrt{s_t}}$$

$$- \frac{\beta_{1,t}}{1 - \beta_{1,t}}\Big(\frac{\alpha_t}{\sqrt{s_t}} - \frac{\alpha_{t-1}}{\sqrt{s_{t-1}}}\Big)m_{t-1} - \frac{\alpha_t g_t}{\sqrt{s_t}}, \forall t > 1 \tag{23}$$

*and*

$$z_2 - z_1 = -\Big(\frac{\beta_{1,2}}{1 - \beta_{1,2}} - \frac{\beta_{1,1}}{1 - \beta_{1,1}}\Big)\frac{\alpha_1 m_1}{\sqrt{v_1}} - \frac{\alpha_1 g_1}{\sqrt{v_1}} \tag{24}$$

**Lemma .6.** *[35] Suppose that the conditions in Theorem (.3) hold, then*

$$\mathbb{E}\Big[f(z_{t+1} - f(z_t))\Big] \leq \sum_{i=1}^{6} T_i \tag{25}$$

*where*

$$T_1 = -\mathbb{E}\Big[\sum_{i=1}^{t}\langle \nabla f(z_i), \frac{\beta_{1,i}}{1 - \beta_{1,i}}\Big(\frac{\alpha_i}{\sqrt{v_i}} - \frac{\alpha_{i-1}}{\sqrt{v_{i-1}}}\Big)m_{i-1}\rangle\Big] \tag{26}$$

$$T_2 = -\mathbb{E}\Big[\sum_{i=1}^{t}\alpha_i\langle \nabla f(z_i), \frac{g_i}{\sqrt{v_i}}\rangle\Big] \tag{27}$$

$$T_3 = -\mathbb{E}\Big[\sum_{i=1}^{t}\langle \nabla f(z_i), \Big(\frac{\beta_{1,i+1}}{1 - \beta_{1,i+1}} - \frac{\beta_i}{1 - \beta_i}\Big)\frac{\alpha_i m_i}{\sqrt{v_i}}\rangle\Big] \tag{28}$$

$$T_4 = \mathbb{E}\Big[\sum_{i=1}^{t}\frac{3L}{2}\Big\|\Big(\frac{\beta_{1,i+1}}{1 - \beta_{1,i+1}} - \frac{\beta_{1,i}}{1 - \beta_{1,i}}\Big)\frac{\alpha_i m_i}{\sqrt{v_i}}\Big\|^2\Big] \tag{29}$$

$$T_5 = \mathbb{E}\Big[\sum_{i=1}^{t}\frac{3L}{2}\Big\|\frac{\beta_{1,i}}{1 - \beta_{1,i}}\Big(\frac{\alpha_i}{\sqrt{v_i}} - \frac{\alpha_{i-1}}{\sqrt{v_{i-1}}}\Big)m_{i-1}\Big\|^2\Big] \tag{30}$$

$$T_6 = \mathbb{E}\Big[\sum_{i=1}^{t}\frac{3L}{2}\Big\|\frac{\alpha_i g_i}{\sqrt{v_i}}\Big\|^2\Big] \tag{31}$$

**Lemma .7.** *[35] Suppose that the condition in Theorem .3 hold, $T_1$ in (26) can be bounded as:*

$$T_1 = -\mathbb{E}\Big[\sum_{i=1}^{t}\langle\nabla f(z_i), \frac{\beta_{1,i}}{1-\beta_{1,i}}\Big(\frac{\alpha_i}{\sqrt{v_i}} - \frac{\alpha_{i-1}}{\sqrt{v_{i-1}}}\Big)m_{i-1}\rangle\Big]$$

$$\leq H^2\frac{\beta_1}{1-\beta_1}\mathbb{E}\Big[\sum_{i=2}^{t}\sum_{j=1}^{d}\Big|\Big(\frac{\alpha_i}{\sqrt{v_i}} - \frac{\alpha_{i-1}}{\sqrt{v_{i-1}}}\Big)_j\Big|\Big] \tag{32}$$

**Lemma .8.** *[35] Suppose the conditions in Theorem .3 are satisfied, then $T_3$ in (28) can be bounded as*

$$T_3 = -\mathbb{E}\Big[\sum_{i=1}^{t}\langle\nabla f(z_i), \Big(\frac{\beta_{1,i+1}}{1-\beta_{1,i+1}} - \frac{\beta_i}{1-\beta_i}\Big)\frac{\alpha_i m_i}{\sqrt{v_i}}\rangle\Big]$$

$$\leq \Big(\frac{\beta_1}{1-\beta_1} - \frac{\beta_{1,t+1}}{1-\beta_{1,t+1}}\Big)(H^2+G^2) \tag{33}$$

**Lemma .9.** *[35] Suppose assumptions in Theorem .3 are satisfied, then $T_4$ in (29) can be bounded as:*

$$T_4 = \mathbb{E}\Big[\sum_{i=1}^{t}\frac{3L}{2}\Big\|\Big(\frac{\beta_{1,i+1}}{1-\beta_{1,i+1}} - \frac{\beta_{1,i}}{1-\beta_{1,i}}\Big)\frac{\alpha_i m_i}{\sqrt{v_i}}\Big\|^2\Big]$$

$$\leq \frac{3L}{2}\Big(\frac{\beta_1}{1-\beta_1} - \frac{\beta_{1,t+1}}{1-\beta_{1,t+1}}\Big)^2 G^2 \tag{34}$$

**Lemma .10.** *[35] Suppose the assumptions in Theorem .3 are satisfied, then $T_5$ in (30) can be bounded as:*

$$T_5 = \mathbb{E}\Big[\sum_{i=1}^{t}\frac{3L}{2}\Big\|\frac{\beta_{1,i}}{1-\beta_{1,i}}\Big(\frac{\alpha_i}{\sqrt{v_i}} - \frac{\alpha_{i-1}}{\sqrt{v_{i-1}}}\Big)m_{i-1}\Big\|^2\Big]$$

$$\leq \frac{3L}{2}\Big(\frac{\beta_1}{1-\beta_1}\Big)^2 H^2\mathbb{E}\Big[\sum_{i=2}^{t}\sum_{j=1}^{d}\Big(\frac{\alpha_i}{\sqrt{v_i}} - \frac{\alpha_{i-1}}{\sqrt{v_{i-1}}}\Big)_j^2\Big] \tag{35}$$

**Lemma .11.** *[35] Suppose the assumptions in Theorem 8 are satisfied, then $T_2$ in (27) are bounded as:*

$$T_2 = -\mathbb{E}\Big[\sum_{i=1}^{t}\alpha_i\langle\nabla f(z_i), \frac{g_i}{\sqrt{v_i}}\rangle\Big]$$

$$\leq \mathbb{E}\sum_{i=2}^{t}\frac{1}{2}\Big\|\frac{\alpha_i g_i}{\sqrt{v_i}}\Big\|^2 + L^2\Big(\frac{\beta_1}{1-\beta_1}\Big)^2\Big(\frac{1}{1-\beta_1}\Big)^2\mathbb{E}\Big[\sum_{j=1}^{d}\sum_{i=2}^{t-1}\Big(\frac{\alpha_i g_i}{\sqrt{v_i}}\Big)_j^2\Big]$$

$$+ L^2 H^2\Big(\frac{\beta_1}{1-\beta_1}\Big)^4\Big(\frac{1}{1-\beta_1}\Big)^2\mathbb{E}\Big[\sum_{j=1}^{d}\sum_{i=2}^{t-1}\Big(\frac{\alpha_i}{\sqrt{v_i}} - \frac{\alpha_{i-1}}{\sqrt{v_{i-1}}}\Big)_j^2\Big]$$

$$+ 2H^2\mathbb{E}\Big[\sum_{j=1}^{d}\sum_{i=2}^{t}\Big|\Big(\frac{\alpha_i}{\sqrt{v_i}} - \frac{\alpha_{i-1}}{\sqrt{v_{i-1}}}\Big)_j\Big|\Big]$$

$$+ 2H^2\mathbb{E}\Big[\sum_{j=1}^{d}\Big(\frac{\alpha_1}{\sqrt{v_1}}\Big)_j\Big]$$

$$- \mathbb{E}\Big[\sum_{i=1}^{t}\alpha_i\langle\nabla f(x_i), \nabla f(x_i)/\sqrt{v_i}\rangle\Big] \tag{36}$$

**Proof of Theorem .3**

 We provide the proof from [35] for completeness. We combine Lemma .5, .6, .7, .8, .9, .10 and .11 to
 bound the objective.

$$\mathbb{E}\Big[f(z_{t+1}) - f(z_t)\Big] \leq \sum_{i=1}^{6} T_i$$

$$\leq H^2 \frac{\beta_1}{1-\beta_1} \mathbb{E}\Bigg[\sum_{i=2}^{t}\sum_{j=1}^{d}\Big|\Big(\frac{\alpha_i}{\sqrt{v_i}} - \frac{\alpha_{i-1}}{\sqrt{v_{i-1}}}\Big)_j\Big|\Bigg]$$

$$+ \Big(\frac{\beta_1}{1-\beta_1} - \frac{\beta_{1,t+1}}{1-\beta_{1,t+1}}\Big)(H^2 + G^2)$$

$$+ \frac{3L}{2}\Big(\frac{\beta_1}{1-\beta_1} - \frac{\beta_{1,t}}{1-\beta_{1,t}}\Big)^2 G^2$$

$$+ \frac{3L}{2}\Big(\frac{\beta_1}{1-\beta_1}\Big)^2 H^2 \mathbb{E}\Bigg[\sum_{i=2}^{t}\sum_{j=1}^{d}\Big(\frac{\alpha_i}{\sqrt{v_i}} - \frac{\alpha_{i-1}}{\sqrt{v_{i-1}}}\Big)_j^2\Bigg]$$

$$+ \mathbb{E}\sum_{i=2}^{t}\frac{1}{2}\Big\|\frac{\alpha_i g_i}{\sqrt{v_i}}\Big\|^2 + L^2\Big(\frac{\beta_1}{1-\beta_1}\Big)^2\Big(\frac{1}{1-\beta_1}\Big)^2 \mathbb{E}\Bigg[\sum_{j=1}^{d}\sum_{i=2}^{t-1}\Big(\frac{\alpha_i g_i}{\sqrt{v_i}}\Big)_j^2\Bigg]$$

$$+ L^2 H^2\Big(\frac{\beta_1}{1-\beta_1}\Big)^4\Big(\frac{1}{1-\beta_1}\Big)^2 \mathbb{E}\Bigg[\sum_{j=1}^{d}\sum_{i=2}^{t-1}\Big(\frac{\alpha_i}{\sqrt{v_i}} - \frac{\alpha_{i-1}}{\sqrt{v_{i-1}}}\Big)_j^2\Bigg]$$

$$+ 2H^2 \mathbb{E}\Bigg[\sum_{j=1}^{d}\sum_{i=2}^{t}\Big|\Big(\frac{\alpha_i}{\sqrt{v_i}} - \frac{\alpha_{i-1}}{\sqrt{v_{i-1}}}\Big)_j\Big|\Bigg]$$

$$+ 2H^2 \mathbb{E}\Bigg[\sum_{j=1}^{d}\Big(\frac{\alpha_1}{\sqrt{v_1}}\Big)_j\Bigg]$$

$$- \mathbb{E}\Bigg[\sum_{i=1}^{t}\alpha_i \langle \nabla f(x_i), \nabla f(x_i)/\sqrt{v_i}\rangle\Bigg]$$

$$\leq \mathbb{E}\Big[C_1 \sum_{t=1}^{T}\Big\|\alpha_t g_t/\sqrt{s_t}\Big\|^2 + C_2 \sum_{t=1}^{T}\Big\|\frac{\alpha_t}{\sqrt{s_t}} - \frac{\alpha_{t-1}}{\sqrt{s_{t-1}}}\Big\|_1$$

$$+ C_3 \sum_{t=1}^{T}\Big\|\frac{\alpha_t}{\sqrt{s_t}} - \frac{\alpha_{t-1}}{\sqrt{s_{t-1}}}\Big\|^2\Big] + C_4 \tag{37}$$

 The constants are defined below:

$$C_1 \triangleq \frac{3}{2}L + \frac{1}{2} + L^2\frac{\beta_1}{1-\beta_1}\Big(\frac{1}{1-\beta_1}\Big)^2 \tag{38}$$

$$C_2 \triangleq H^2\frac{\beta_1}{1-\beta_1} + 2H^2 \tag{39}$$

$$C_3 \triangleq \Big[1 + L^2\Big(\frac{1}{1-\beta_1}\Big)^2\Big(\frac{\beta_1}{1-\beta_1}\Big)\Big]H^2\Big(\frac{\beta_1}{1-\beta_1}\Big)^2 \tag{40}$$

$$C_4 \triangleq \Big(\frac{\beta_1}{1-\beta_1}\Big)(H^2 + G^2) + \Big(\frac{\beta_1}{1-\beta_1}\Big)^2 G^2 + 2H^2\mathbb{E}\big[||\alpha_1/\sqrt{v_1}||_1\big] + \mathbb{E}[f(z_1) - f(z^*)] \tag{41}$$

 $\qquad\qquad\qquad\qquad\qquad\qquad\qquad\qquad\qquad\qquad\qquad\qquad\qquad\qquad\qquad\qquad\qquad\qquad\qquad\quad$ □

# E. Bayesian interpretation of AdaBelief

 We analyze AdaBelief from a Bayesian perspective.

647 **Theorem .12.** *Assume the gradient follows a Gaussian prior with uniform diagonal covariance,*
648 $\tilde{g} \sim \mathcal{N}(0, \sigma^2 I)$; *assume the observed gradient follows a Gaussian distribution,* $g \sim \mathcal{N}(\tilde{g}, C)$, *where*
649 $C$ *is some covariance matrix. Then the posterior is:* $\tilde{g}|g, C \sim \mathcal{N}\left((I + \frac{C}{\sigma^2})^{-1} g, (\frac{I}{\sigma^2} + C^{-1})^{-1}\right)$

650 We skip the proof, which is a direct application of the Bayes rule in the Gaussian distribution case as
651 in [60]. If $g$ is averaged across a batch of size $n$, we can replace $C$ with $\frac{C}{n}$.

652 According to Theorem .12, the gradient descent direction with maximum expected gain is:

$$\mathbb{E}\left[\tilde{g}|g, C\right] = (I + \frac{C}{\sigma^2})^{-1} g = \sigma^2 (\sigma^2 I + C)^{-1} g \propto (\sigma^2 I + C)^{-1} g \tag{42}$$

653 Denote $\epsilon = \sigma^2$, then adaptive optimizers update in the direction $(\epsilon I + C)^{-1} g$; considering the
654 noise in $g_t$, in practice most optimizers replace $g_t$ with its EMA $m_t$, hence the update direction is
655 $(\epsilon I + C)^{-1} m_t$. In practice, adaptive methods such as Adam and AdaGrad replace $(\epsilon I + C)^{-1/2}(\epsilon I +$
656 $C)^{-1/2} m_t$ with $\alpha I (\epsilon I + C)^{-1/2} m_t$ for numerical stability, where $\alpha$ is some predefined learning
657 rate. Both Adam and AdaBelief take this form; their difference is in the estimate of $C$: Adam
658 uses an *uncentered* approximation $C_{Adam} \approx \text{EMA diag}(g_t g_t^\top)$, while AdaBelief uses a *centered*
659 approximation $C_{AdaBelief} \approx \text{EMA diag}[(g_t - \mathbb{E}g_t)(g_t - \mathbb{E}g_t)^\top]$. Note that the definition of $C$ is
660 the *covariance* hence it is *centered*. Note that for the $i$th parameter, $\mathbb{E}(g_t^i)^2 = (\mathbb{E}g_t^i)^2 + \text{Var}(g_t^i)$, so
661 when $\text{Var} \, g_t^i \ll ||\mathbb{E}g_t^i||$, we have $C_{AdaBelief}^i < C_{Adam}^i$, and AdaBelief behaves closer to the ideal
662 and takes a larger step than Adam because $C$ is in the denominator.

663 From a practical perspective, $\epsilon$ can be interpreted as a numerical term to avoid division by 0; from the
664 Bayesian perspective, $\epsilon$ represents our prior on $g_t$, with a larger $\epsilon$ indicating a larger $\sigma^2$. Note that
665 as the network evolves with training, the distribution of the gradient is distorted (an example with
666 Adam is shown in Fig. 2 of [16]), hence the Gaussian prior might not match the true distribution. To
667 solve the mismatch between prior and the true distribution, it might be reasonable to use a weak prior
668 during late stages of training (e.g., let $\sigma^2$ grow at late training phases, and when $\sigma^2 \to \infty$ reduces to
669 a uniform prior). We only provide a Bayesian perspective here, and leave the detailed discussion to
670 future works.

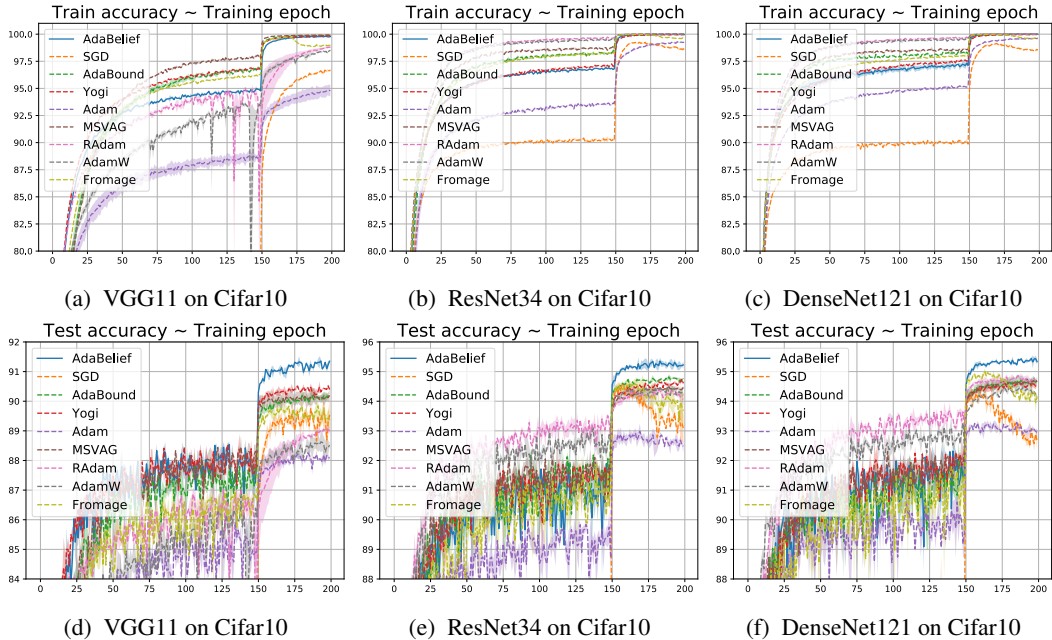

Figure 1: Training (top row) and test (bottom row) accuracy of CNNs on Cifar10 dataset. We report confidence interval $[\mu \pm \sigma]$ of 3 independent runs.

## F. Experimental Details

### 1. Image classification with CNNs on Cifar

We performed experiments based on the official implementation[3] of AdaBound [12], and exactly replicated the results of AdaBound as reported in [12]. We then experimented with different optimizers under the same setting: for all experiments, the model is trained for 200 epochs with a batch size of 128, and the learning rate is multiplied by 0.1 at epoch 150. We performed extensive hyperparameter search as described in the main paper. In the main paper we only report test accuracy; here we report both training and test accuracy in Fig. 1 and Fig. 2. AdaBelief not only achieves the highest test accuracy, but also a smaller gap between training and test accuracy compared with other optimizers such as Yogi.

### 2. Image Classification on ImageNet

We experimented with a ResNet18 on ImageNet classcation task. For SGD, we use the same learning rate schedule as [25], with an initial learning rate of 0.1, and multiplied by 0.1 at epoch 30 and 60; for AdaBelief, we use an initial learning rate of 0.001, and decayed it at epoch 70 and 80. Weight decay is set as $10^{-4}$ for both cases. To match the settings in [?] and [16], we use decoupled weight decay. As shown in Fig. 3, AdaBelief achieves an accuracy very close to SGD, closing the generalization gap between adaptive methods and SGD. Meanwhile, when trained with a large learning rate (0.1 for SGD, 0.001 for AdaBelief), AdaBelief achieves faster convergence than SGD in the initial phase.

### 3. Robustness to hyperparameters

**Robustness to $\epsilon$**   We test the performances of AdaBelief and Adam with different values of $\epsilon$ varying from $10^{-4}$ to $10^{-9}$ in a log-scale grid. We perform experiments with a ResNet34 on Cifar10 dataset, and summarize the results in Fig. 4. Compared with Adam, AdaBelief is slightly more sensitive to the choice of $\epsilon$, and achieves the highest accuracy at the default valiue $\epsilon = 10^{-8}$; AdaBelief achieves accuracy higher than 94% for all $\epsilon$ values, consistently outperforming Adam which achieves an accuracy around 93%.

---

[3]`https://github.com/Luolc/AdaBound`

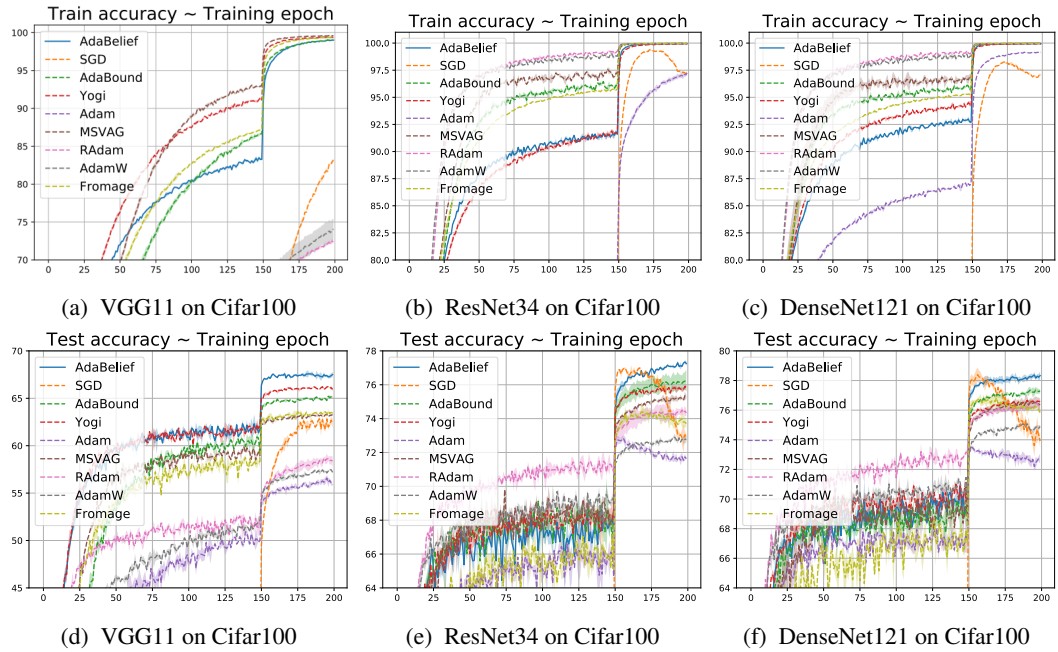

(a) VGG11 on Cifar100     (b) ResNet34 on Cifar100     (c) DenseNet121 on Cifar100

(d) VGG11 on Cifar100     (e) ResNet34 on Cifar100     (f) DenseNet121 on Cifar100

Figure 2: Training (top row) and test (bottom row) accuracy of CNNs on Cifar10 dataset. We report confidence interval $[\mu \pm \sigma]$ of 3 independent runs.

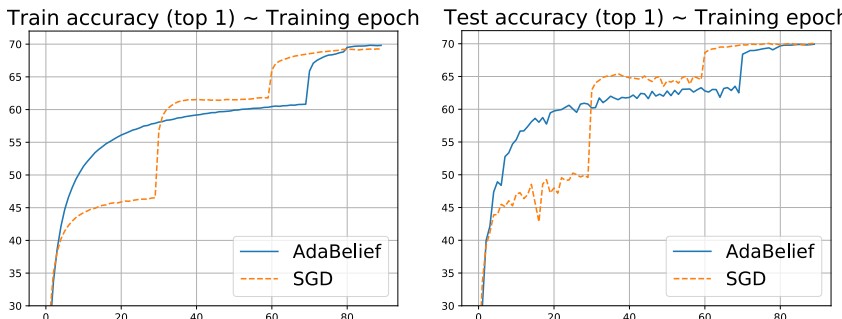

Figure 3: Training and test accuracy (top-1) of ResNet18 on ImageNet.

**Robustness to learning rate** We test the performance of AdaBelief with different learning rates. We experiment with a VGG11 network on Cifar10, and display the results in Fig. 5. For a large range of learning rates from $5 \times 10^{-4}$ to $3 \times 10^{-3}$, compared with Adam, AdaBelief generates higher test accuracy curve, and is more robust to the change of learning rate.

## 4. Experiments with LSTM on language modeling

We experiment with LSTM models on Penn-TreeBank dataset, and report the results in Fig. 6. Our experiments are based on this implementation [4]. Results $[\mu \pm \sigma]$ are measured across 3 runs with independent initialization. For completeness, we plot both the training and test curves.

We use the default parameters $\alpha = 0.001, \beta_1 = 0.9, \beta_2 = 0.999, \epsilon = 10^{-8}$ for 2-layer and 3-layer models; for 1-layer model we set $\epsilon = 10^{-12}$ and set other parameters as default. For simple models (1-layer LSTM), AdaBelief's perplexity is very close to other optimizers; on complicated models, AdaBelief achieves a significantly lower perplexity on the test set.

---

[4] https://github.com/salesforce/awd-lstm-lm

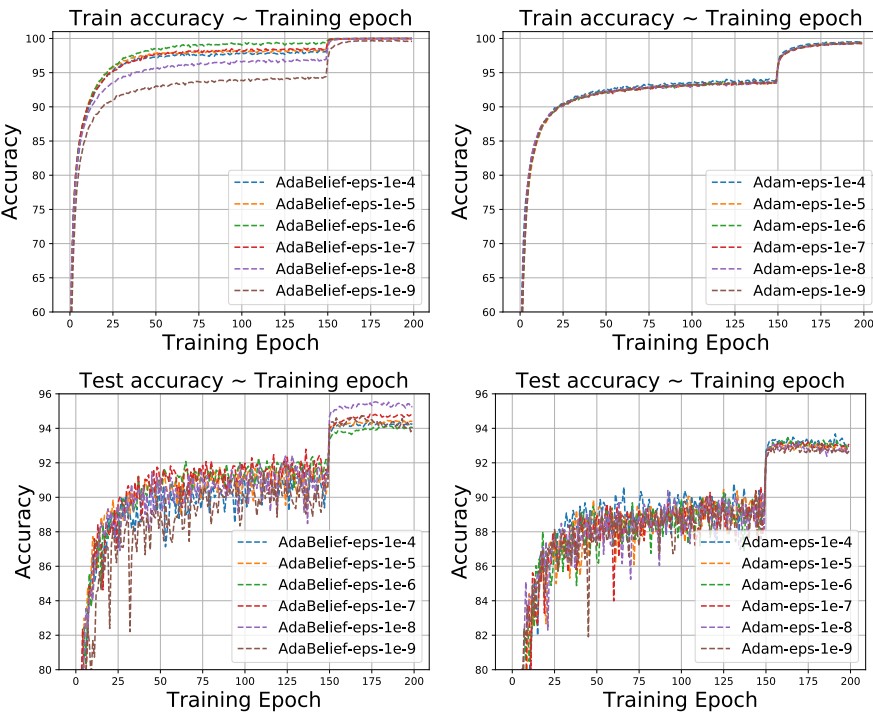

Figure 4: Training (top row) and test (bottom row) accuracy of ResNet34 on Cifar10, trained with AdaBelief (left column) and Adam (right column) using different values of $\epsilon$. Note that AdaBelief achieves an accuracy above $94\%$ for all $\epsilon$ values, while Adam's accuracy is consistently below $94\%$.

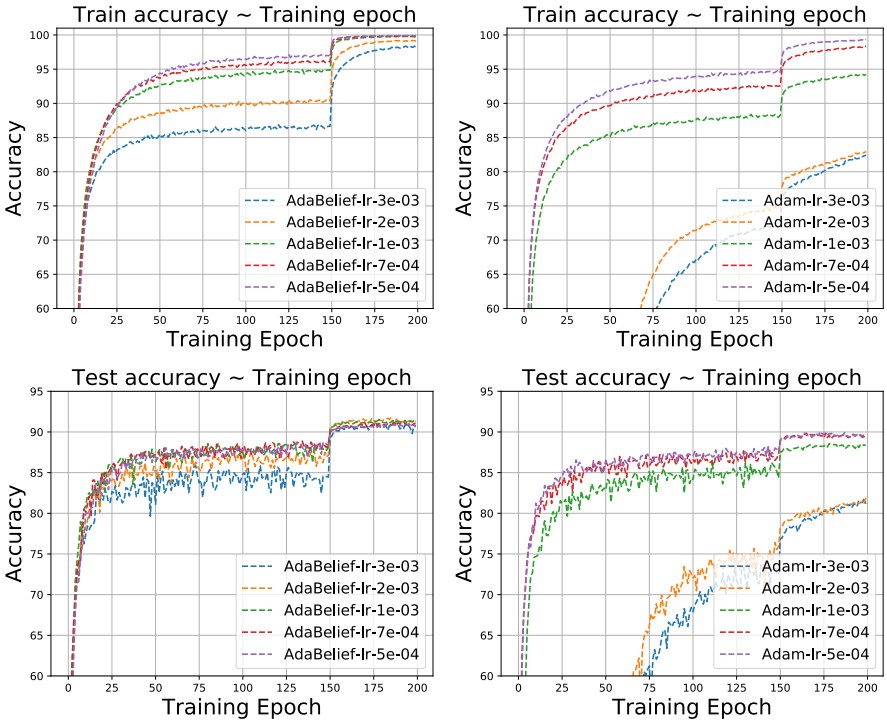

Figure 5: Training (top row) and test (bottom row) accuracy of VGG on Cifar10, trained with AdaBelief (left column) and Adam (right column) using different values of learning rate.

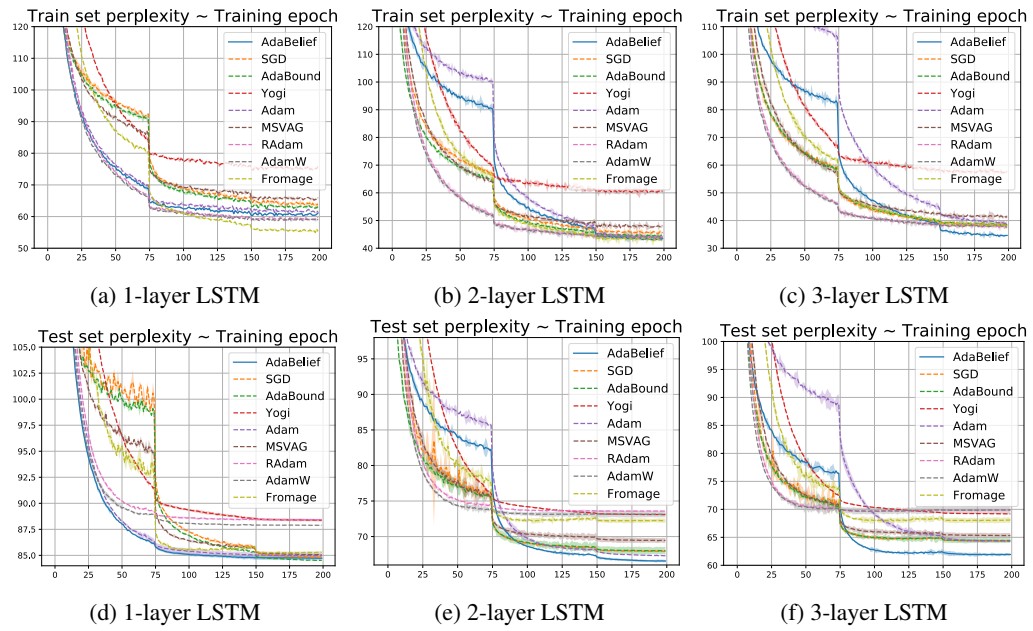

Figure 6: Training (top row) and test (bottom row) perplexity on Penn-TreeBank dataset, lower is better.

Table 1: Structure of GAN

| Generator | Discriminator |
|---|---|
| ConvTranspose ([inchannel = 100, outchannel = 512, kernel = 4×4, stride = 1]) | Conv2D([inchannel=3, outchannel=64, kernel = 4×4, stride=2]) |
| BN-ReLU | LeakyReLU |
| ConvTranspose ([inchannel = 512, outchannel = 256, kernel = 4×4, stride = 2]) | Conv2D([inchannel=64, outchannel=128, kernel = 4×4, stride=2]) |
| BN-ReLU | BN-LeakyReLU |
| ConvTranspose ([inchannel = 256, outchannel = 128, kernel = 4×4, stride = 2]) | Conv2D([inchannel=128, outchannel=256, kernel = 4×4, stride=2]) |
| BN-ReLU | BN-LeakyReLU |
| ConvTranspose ([inchannel = 128, outchannel = 64, kernel = 4×4, stride = 2]) | Conv2D([inchannel=256, outchannel=512, kernel = 4×4, stride=2]) |
| BN-ReLU | BN-LeakyReLU |
| ConvTranspose ([inchannel = 64, outchannel = 3, kernel = 4×4, stride = 2]) | Linear(-1, 1) |
| Tanh | |

## 5. Experiments with GAN

We experimented with a WGAN [30] and WGAN-GP [39]. The code is based on several public github repositories [5],[6]. We summarize network structure in Table 1. For WGAN, the weight of discriminator is clipped within $[-0.01, 0.01]$; for WGAN-GP, the weight for gradient-penalty is set as 10.0, as recommended by the original implementation. For each optimizer, we perform 5 independent runs. We train the model for 100 epochs, generate 64,000 fake samples (60,000 real images in Cifar10), and measure the Frechet Inception Distance (FID) [40] between generated samples and real samples. Our implementation on FID heavily relies on an open-source implementation[7]. We report the FID scores in the main paper, and demonstrate fake samples in Fig. 7 and Fig. 8 for WGAN and WGAN-GP respectively.

---

[5] https://github.com/pytorch/examples

[6] https://github.com/eriklindernoren/PyTorch-GAN

[7] https://github.com/mseitzer/pytorch-fid

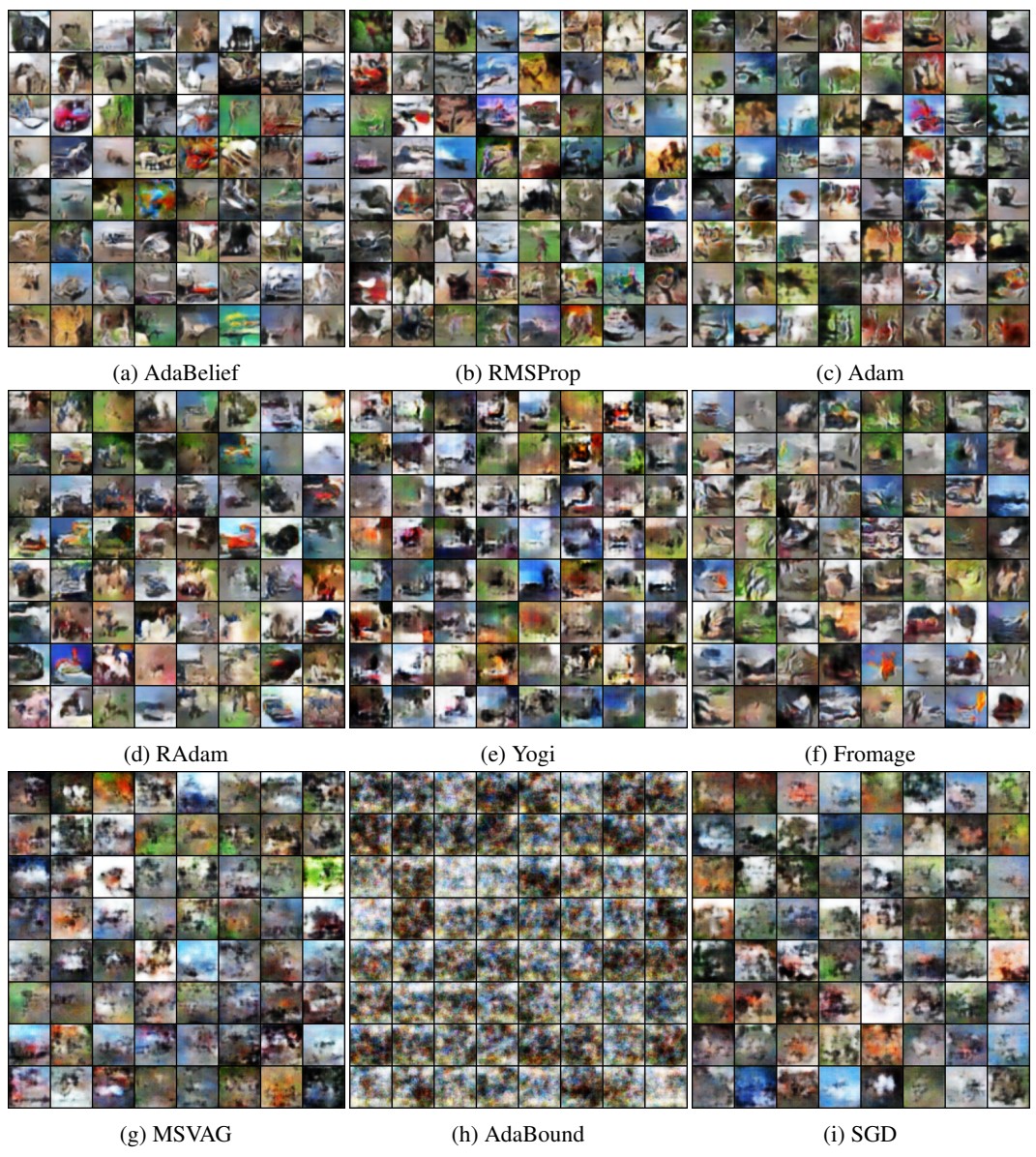

(a) AdaBelief       (b) RMSProp       (c) Adam

(d) RAdam       (e) Yogi       (f) Fromage

(g) MSVAG       (h) AdaBound       (i) SGD

Figure 7: Fake samples from WGAN trained with different optimizers.

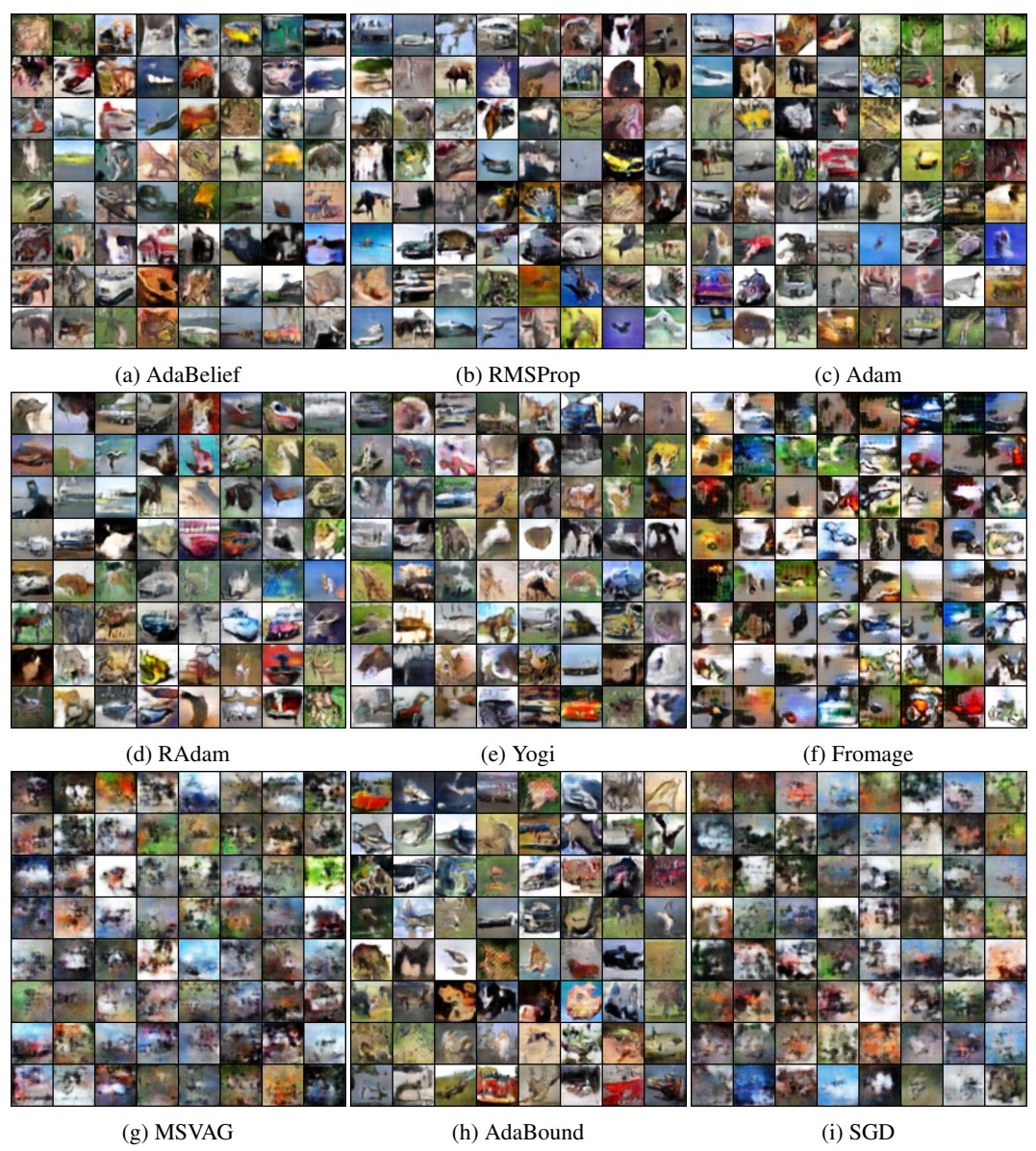

(a) AdaBelief        (b) RMSProp        (c) Adam

(d) RAdam        (e) Yogi        (f) Fromage

(g) MSVAG        (h) AdaBound        (i) SGD

Figure 8: Fake samples from WGAN-GP trained with different optimizers.

