# OpenReview forum: "AdaBelief Optimizer: Adapting Stepsizes by theBelief in Observed Gradients"
_NeurIPS.cc/2020/Workshop/DL-IG — NeurIPSW 2020: DL-IG Poster_

### Official Review · AnonReviewer1 · 2020-11-01

**Rating:** 7
**Confidence:** 4

**Review:**

This paper presents an algorithm for training deep networks called AdaBelief. The algorithm is a modification of Adam and changes the how the run-time average of the second moment of the gradient is calculated. The motivation for this modification comes through some simple yet revealing examples in Section 2.2; AdaBelief can be heuristically connected to second-order updates like those in Newton’s method and while Adam is essentially sign-SGD for low-variance gradients, AdaBelief considers both the sign and the magnitude of the gradient.

Empirical  results are provided on a large number of tasks from classification, language modeling and generative modeling. The results on ImageNet are impressive.

---

### Official Review · AnonReviewer2 · 2020-11-03
**Review on "AdaBelief Optimizer: Adapting Stepsizes by theBelief in Observed Gradients"**

**Rating:** 9
**Confidence:** 4

**Review:**

The paper is about a proposed modification of the Adam optimization which is claimed to have fast convergence, comparable generalization of the trained models to those obtained using SGD, and good training stability. The basic idea behind this improvement is to adapt the stepsize according to the so-called "belief" in the current gradient direction (i.e., if the gradients largely deviate from the exponential moving average, then take small steps, otherwise continue taking larger steps). The method is shown to work efficiently in multiple benchmark problems. While I recommend the authors to test the method on wider types of landscapes (e.g., benchmark optimization problems in 2 and 3 dimensions), I recommend to accept the paper for the workshop.

---

### Decision · Program_Chairs · 2020-11-07

Accept (Poster)